# Novel Instance-Based Transfer Learning for Asphalt Pavement Performance Prediction

Jiale Li [1,*], Jiayin Guo [1], Bo Li [2] and Lingxin Meng [3,4]

1   School of Civil and Transportation Engineering, Hebei University of Technology, Tianjin 300401, China; 202131604014@stu.hebut.edu.cn
2   Wenzhou Key Laboratory of Intelligent Lifeline Protection and Emergency Technology for Resilient City, Wenzhou University of Technology, Wenzhou 325035, China; autumnlibo@gmail.com
3   School of Civil Engineering, Zhejiang University of Technology, Hangzhou 310014, China; 111122060011@zjut.edu.cn
4   Wenzhou Xinda Transportation Engineering Testing Co., Ltd., Wenzhou 325105, China
*   Correspondence: jiale.li@hebut.edu.cn

**Abstract:** The deep learning method has been widely used in the engineering field. The availability of the training dataset is one of the most important limitations of the deep learning method. Accurate prediction of pavement performance plays a vital role in road preventive maintenance (PM) and decision-making. Pavement performance prediction based on deep learning has been widely used around the world for its accuracy, robustness, and automation. However, most of the countries in the world have not built their pavement performance historical database, which prevents preventive maintenance using the deep learning method. This study presents an innovative particle swarm optimization (PSO) algorithm-enhanced two-stage TrAdaBoost.$R^2$ transfer learning algorithm, which could significantly increase the pavement performance prediction database. The Long-Term Pavement Performance (LTPP) database is used as the source domain data, and one of the highways in China is chosen as the target domain to predict pavement performance. The results show that the proposed PSO-Two-stage TrAdaBoost.$R^2$ model has the highest accuracy compared with AdaBoost.$R^2$ model and traditional regression decision tree model. The validation case study shows significant consistency between the predicted International Roughness Index (IRI) and the whole-year measurement data with an $R^2$ of 0.7. This study demonstrates the great potential of the innovative instance-based transfer learning method in pavement performance prediction of a region's lack of data. This study also contributes to other engineering fields that could greatly increase the universality of deep learning.

**Keywords:** instance-based transfer learning; preventive maintenance; long-term pavement performance (LTPP); PSO-Two-stage TrAdaBoost.$R^2$

## 1. Introduction

The deep learning method has been proven to be an effective method and has been widely used in solving many engineering issues. One of the main disadvantages of deep learning is that it depends on a large database. Some regions in the world, especially developing countries, have not built their engineering database, which prevents them from using the deep learning method. This study provides an alternative idea to take advantage of the established open-source database by using an instance-based transfer learning algorithm in the engineering field. The total mileage of roads in China reached 5.28 million km by the end of 2021, and in the United States, it reached 6.657 million km by the end of 2020, according to the Ministry of Transpiration of China and the U.S. Department of Transportation (DOT). Road surfaces will exhibit a variety of damages, and maintenance is among the most important tasks of the road management department [1]. Timely preventive maintenance needs to be taken, which means making minor repairs

before roads are seriously damaged [2]. Various studies have presented different artificial neural network (ANN) models for predicting pavement performance recently. However, the ANN method usually depends on big data to achieve acceptable accuracy. Most of the pavement in the U.S. has been built over 50 years, and the U.S. DOT established the Long-Term Pavement Performance (LTPP) database in the 1980s [3]. On the other hand, the majority of roads in China have been built during the past 20 years, and there is no network-level historical database. This situation can be extended to other developing countries that lack pavement performance historical databases [4]. It is of great importance to develop a method to take advantage of the existing database to predict the pavement performance of new roads.

## 2. Literature Review

Pavement deterioration is a complex process determined by many factors, which makes it difficult to predict precisely [5,6]. Pavement performance prediction models are mainly divided into deterministic and probabilistic prediction models in traditional road performance prediction. Deterministic models include the mechanical model, empirical model, and mechanical–empirical model [7]. The difference is that the mechanical model is based on physical and mechanical principles [8], while the empirical model is based on statistical interpretation of observed field performance. Chen et al. [9] determined the average stress state of the foundation layer according to the mechanical equation. They incorporated the effect of load distribution into the model and proposed a new mechanical empirical fault model. Dong et al. [10] combined two influencing factors and established a multi-influencing factor model to analyze the influence of structural equation modeling (SEM). Bayesian and Markov probabilistic prediction models are the main components of probabilistic models. Khaled et al. [11] proposed a discrete-time Markov model based on inverse calculation, achieving a prediction model for cost reduction. Khawaga et al. [12] developed Markov chain-based and sigmoidal curve-based models to predict IRI. The results show that the Markov chain-based model is better than the sigmoidal curve-based model across comprehensive factors. Yang et al. [13] used the dynamic Markov model to predict pavement cracking performance. The results show that the dynamic Markov chain has higher prediction accuracy than the static Markov chain.

In recent years, artificial intelligence (AI) methods have been widely used in road performance prediction for their strong nonlinear fitting abilities, complex theoretical derivation, and real-time prediction capabilities. ANN has great potential when dealing with large amounts of historical data. ANN has better performance than linear regression in predicting IRI using the same stable data source [14]. Sollazzoa et al. [15] used ANN to verify the relationship between roughness and structural performance of asphalt pavement. Gong et al. [16] used the ANN method to predict pavement fatigue cracking. The results showed that the performance of ANN was better than that of the fatigue cracking (FC) transfer function, and the prediction performance of the models with multiple input variables was much better than the models with only two input variables. Li et al. [17] established a hybrid neural network model based on ten years of historical data from a highway in China. The proposed model could predict pavement performance precisely compared with traditional ANN methods. Liu et al. [18] proposed a Mask region-based convolutional neural network model for the automatic detection and segmentation of small cracks in asphalt pavement at the pixel level. Angela J. Haddad et al. [19] obtained a rutting prediction model based on rutting data extracted from the LTPP database using a deep neural network model for training, and the rutting prediction model showed stronger prediction ability compared with commonly used models, and a sensitivity analysis was performed. In addition, generic rutting prediction curves were developed to make rutting predictions available for all road agencies. Wang et al. [20], using the ResNeXt101 network, balanced feature pyramid (BFP), and deformable convolutional (DCN), modified the Hybrid Task Cascade (HTC), proposing an improved hybrid task cascade instance segmentation model that accurately segments pavement surface distress. In addition to

ANN methods, many other machine learning algorithms also have good performance in road performance prediction. Freund and Schapire [21] proposed the AdaBoost model based on the boosting algorithm in 1997, which is one of the most widely used algorithms. Wang et al. [22] used the AdaBoost regression model to predict IRI and showed good results compared with linear regression. The roughness of asphalt concrete pavement is caused by many factors, such as cracks, ruts, and looseness, which contain both linear and nonlinear relationships. The AdaBoost algorithm has the advantage of finding these relationships through complex calculations, which provides the possibility for its application in IRI modeling prediction.

The United States started to build the Long-Term Pavement Performance (LTPP) database in the 1980s. The LTPP database contains a large amount of road historical data, which is of great importance in the study of road-related factors and the development of road performance prediction models. Many studies have used the LTPP database to study pavement performance. Abdelaziz et al. [14] established a prediction model of IRI based on the LTPP database, using both regression models and artificial neural networks to evaluate the structural bearing capacity of pavements. Madeh et al. [23] built five machine learning models to study the impact of climate change on pavement performance. Gong et al. [24] used a random forest regression model to predict the IRI of pavements based on the LTPP database.

One of the primary limitations of machine learning models is the size of the dataset. In the above-mentioned studies, the machine learning models are all based on a large amount of road historical data. However, not all roads have a large amount of historical data, especially those that are newly constructed. It is impossible to build accurate prediction models to provide guidance to road maintenance agencies. Therefore, it is necessary to investigate methods that can predict pavement performance based on the current amount of data [4]. Traditional machine learning methods usually assume that the data generation mechanism does not change with the environment and that the source domain data and the target domain data are required to have the same distribution. The emergence of transfer learning breaks this limitation. It solves the problems of insufficient training datasets, low model classification, and recognition accuracy in machine learning [25]. As long as there is a certain relationship between the source domain and the target domain, the knowledge that has been extracted from the source domain data and features can be used in training the target domain classification model [26], enabling the reuse and transfer of learned knowledge between similar or related fields, transitioning traditional learning from scratch into cumulative learning. Zhang et al. [27] applied deep convolutional neural networks based on transfer learning for model pre-training to classify road images into three types. The experimental results show that the method can successfully distinguish different types of pavement damage. Jang et al. [28] proposed a pre-trained deep convolutional network model based on ImageNet, combined with the transfer learning method, and fine-tuned the model to improve training accuracy and reduce time overhead. Similarly, Gopalakrishnan et al. [29] proposed using a pre-trained deep convolutional neural network model and transfer learning method for automatic pavement crack detection. It is proven that this transfer learning method is successful in pavement crack recognition and detection. Other scholars have also successfully explored the application of transfer learning in other situations, such as visual-based automatic detection of concrete surface cracks [30–32]. Most of the transfer learning applications in road engineering utilize image recognition technology, such as image-based pavement damage detection. Although transfer learning techniques are the most common applications in the field of image analysis, these techniques can be applied to any situation where target domain data is scarce and data from different datasets (source data) are available. Marcelino et al. [4] proposed a transfer learning method based on the Boosting algorithm to predict pavement performance in a limited data environment. The study used the TrAdaBoost algorithm to predict pavement performance using both the LTPP database and the Portuguese road database. However, their study

chose separate sections of the road to make predictions, which makes it hard for the maintenance department to use.

In addition, a few studies have attempted to use transfer learning in other engineering fields. Tang et al. [33] used road crash data from two Chinese cities, Shanghai and Guangzhou, as source and target data domains, using the TrAdaBoost.R$^2$ algorithm to test the adaptability of transfer learning to small samples. Their results show that the model constructed with TrAdaBoost.R$^2$ has better performance than traditional calibration methods. Ahmad et al. [34] used the two-stage TrAdaBoost.R$^2$ algorithm to study knowledge acquisition from available source domain data to predict crash efficiency in the target domain. Their results show that the boosting correction technique has better prediction accuracy than the NB-based correction model with limited target area data. Lv et al. [35] proposed a semi-supervised transfer learning approach for air quality assessment in non-urban areas without air quality monitoring stations. Their results show that the strategy of distinguishing between urban and non-urban areas and combining transfer and semi-supervised learning is effective for air quality assessment. Chen et al. [36] proposed a new TrAdaBoost-LSTM algorithm to use relevant knowledge from complete datasets to increase prediction accuracy of low-quality datasets by 15% to 25%.

Overall, previous studies in other engineering fields have been based on a small number of local databases and did not optimize hyperparameters. Most of the previous studies made pavement performance predictions based on big historical databases, and few studies used transfer learning methods for actual road performance prediction. In addition, many studies have used data from the same period to train and verify the model, lacking temporal prediction. In practice, however, the management department is eager to predict the pavement performance of specific locations in the future. This study predicts, for the first time, the road International Roughness Index (IRI) for a specific road at 100 m intervals for the next year only based on a small amount of data using the instance-based transfer learning method. The research framework is shown in Figure 1.

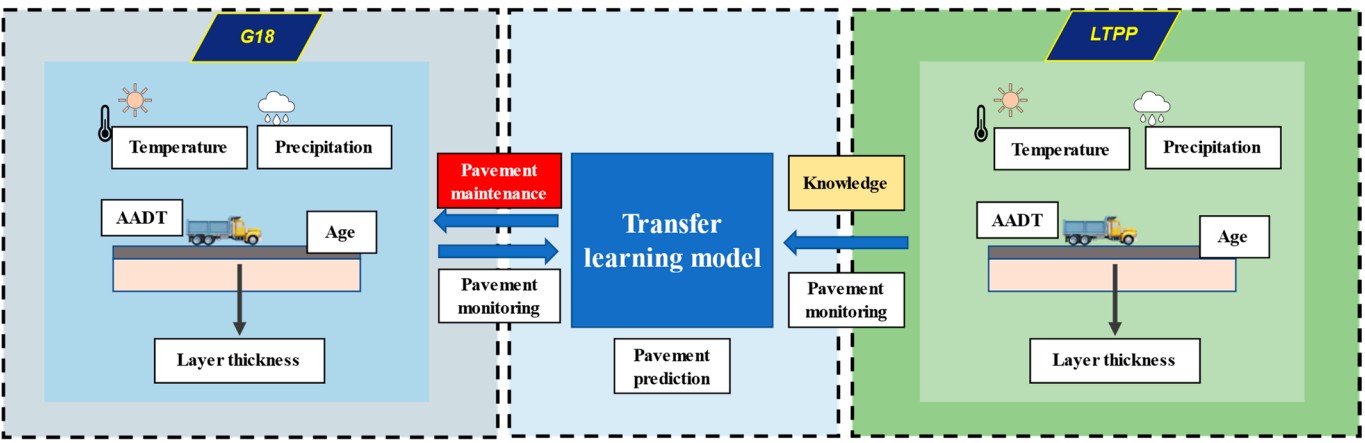

**Figure 1.** Research framework.

The structure of the paper is organized as follows: first, the research background and literature are introduced. Secondly, the test source data and target data are presented. Section 4 introduces the methods and algorithms used in this study. Section 5 presents a detailed analysis and shows the results. Finally, conclusions and summaries are made in Section 6.

## 3. Data Preparation

### 3.1. Long-Term Pavement Performance (LTPP) Program

The Long-Term Pavement Performance (LTPP) database is the world's largest and most comprehensive pavement performance database. The LTPP program aims to study how various road structural thicknesses, climatic conditions, traffic loads, materials, and

maintenance practices affect pavement performance [24]. It is managed by the Federal Highway Administration (FHWA) and includes pavement performance data for more than 2500 road sections. Inventory, maintenance, monitoring, repair, material testing, traffic, and climate data of different types are stored in seven modules of the database. The LTPP program includes two main components: General Pavement Studies (GPS) and Specific Pavement Studies (SPS) the IRI geographical distribution map of the LTPP database shown in Figure 2. The Circles of different colors and sizes in the upper right of the map represent IRI indexes of different sizes, and the numbers in the circles in the map represent the number of road sections in the location. In general, each LTPP test section is 152 m long, each lane is 3.65 m, and all sections are monitored using the same standards. One of the main purposes of the LTPP program is to provide high-quality data for research, and a lot of studies have used the LTPP databases to understand the performance of various maintenance and repair strategies [37–40] and optimize maintenance decision-making processes [41]. The transfer learning conducted in this study requires a large amount of road-related source domain data, and most of the data in the LTPP database are well-observed. Therefore, the international roughness index (IRI), climate data, traffic volume, and road structure information of asphalt pavement from the LTPP database are chosen as the source domain data for this study. In this study, 2611 sets of LTPP data are used as part of the source domain.

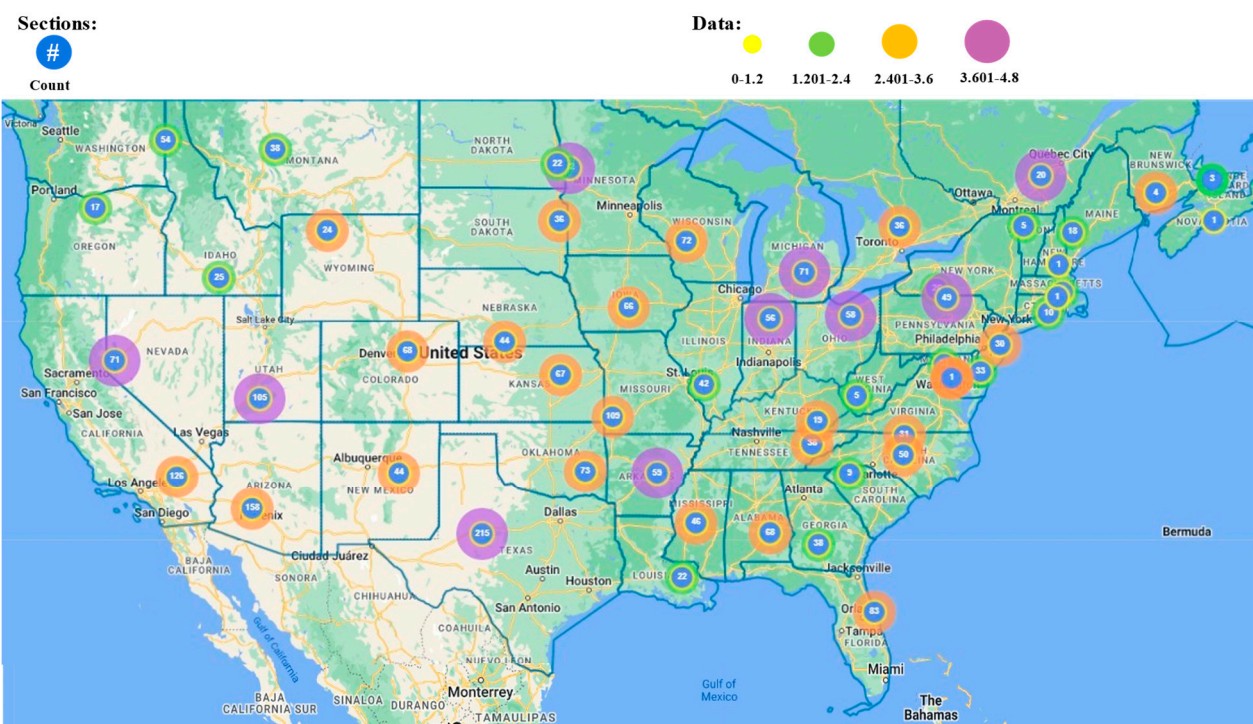

**Figure 2.** The IRI geographical distribution map of the LTPP database (https://infopave.fhwa.dot.gov/Media/LTPPSectionMapping, accessed on 25 December 2023).

*3.2. Target Data Source*

The experimental data used in this study pertain to a part of China's G18 national highway. The road pavement is asphalt concrete pavement [17]. The highway has three lanes in both directions, as shown in Figure 3. To separate the lanes in both directions, the lanes are named the upward direction and the downward direction. As part of the data on the first lane are unavailable, the second and third lanes in both directions are considered in this study. The structure of the pavement consists of the following layers: 4 cm of fine-grained modified asphalt concrete (AC-13C), 6 cm of medium-grained modified asphalt concrete (AC-20C), 8 cm of coarse-grained asphalt

concrete (AC-25C), 18 cm of cement stabilized macadam, 18 cm of cement-stabilized macadam, and 18 cm of lime fly ash soil. Although there are many differences between Chinese standards and U.S. standards, the two nations use the same International Roughness Index to evaluate the roughness of the pavement, according to the Highway Performance Assessment Standards of China [42]. Therefore, this study chooses the international roughness index (IRI) as the research target value.

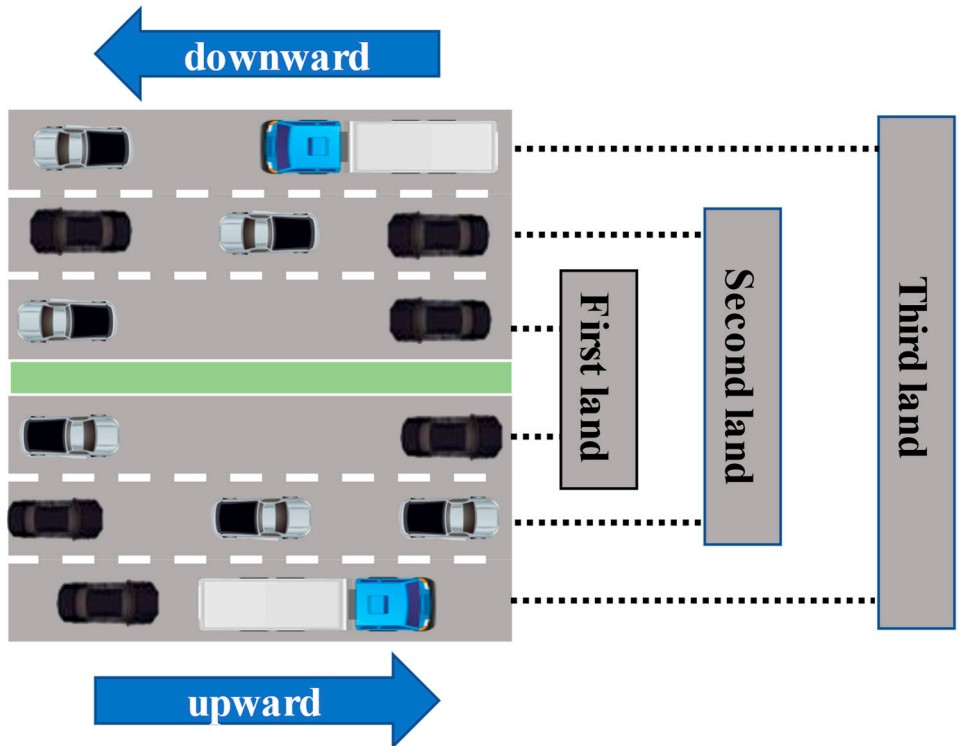

**Figure 3.** Schematic diagram of lands.

An intelligent road condition detection vehicle is used to measure road roughness. It uses a flatness detection system within the vehicle to detect the road surface flatness and measure the International Roughness Index (IRI) of the road in real-time. The sampling interval is 10 cm, and test results are summarized every 100 m to evaluate road surface flatness. The traffic data of the highway are represented by the annual average daily traffic volume (AADT). The climate data, including the annual average temperature and the annual total rainfall, are provided by the weather station,. In addition, road structure data are divided into thickness of the asphalt layer and the road base.

In this study, data from the four lanes of the G18 highway were utilized as the target domain, and each lane was used for transfer learning. The second and third lanes served as the target domain in model development. After data cleaning, the second lane upward had 202 sets of data, the second lane downward had 199 sets of data, the third lane upward had 306 sets of data, and the third lane downward had 309 sets of data. Figure 4 shows the data distribution of the international roughness index (IRI) from both the LTPP database and the G18 database, respectively. It can be seen from the diagram that the distribution of the IRI in the two databases takes the form of a skewed normal distribution. The data are mostly distributed around 1.0 for the LTPP database and 1.5 for the G18 dataset. This may be due to the G18 having a big ratio of heavy traffic, resulting in a large IRI. On the other hand, the two databases have a certain similarity, which should be helpful for subsequent data migration.

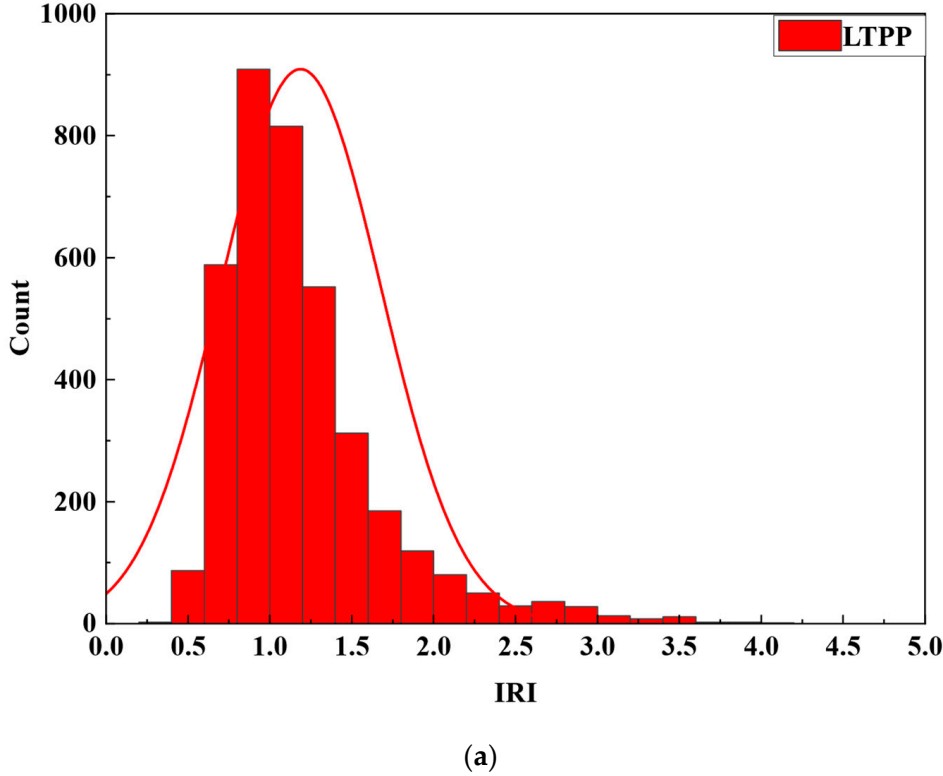

(**a**)

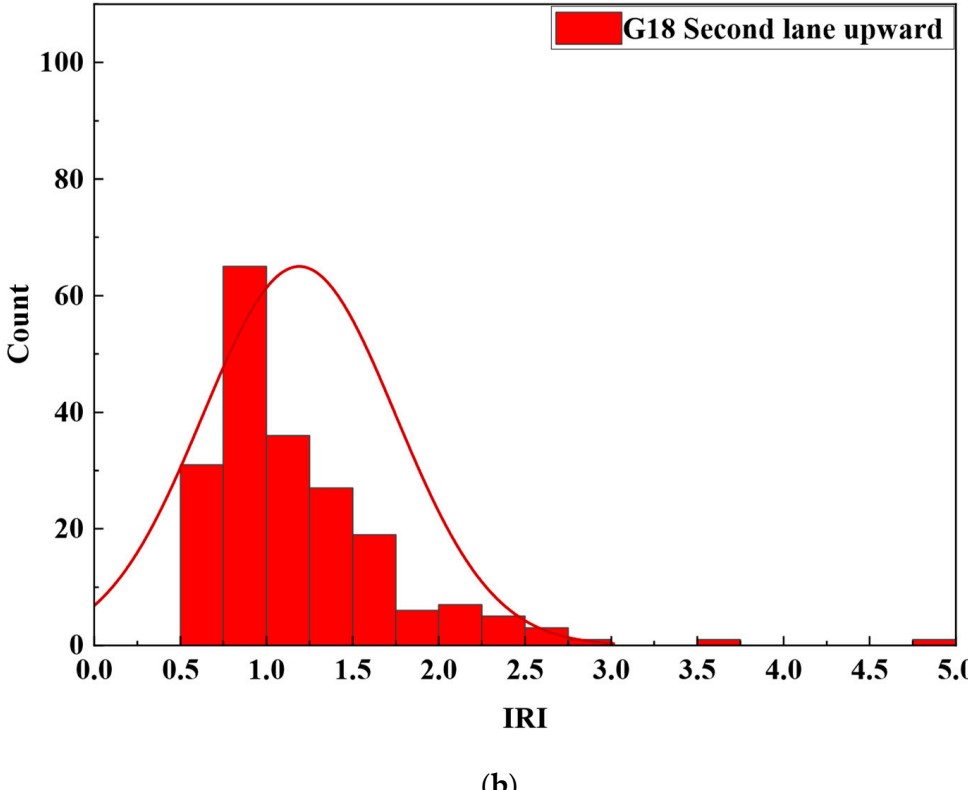

(**b**)

**Figure 4.** *Cont.*

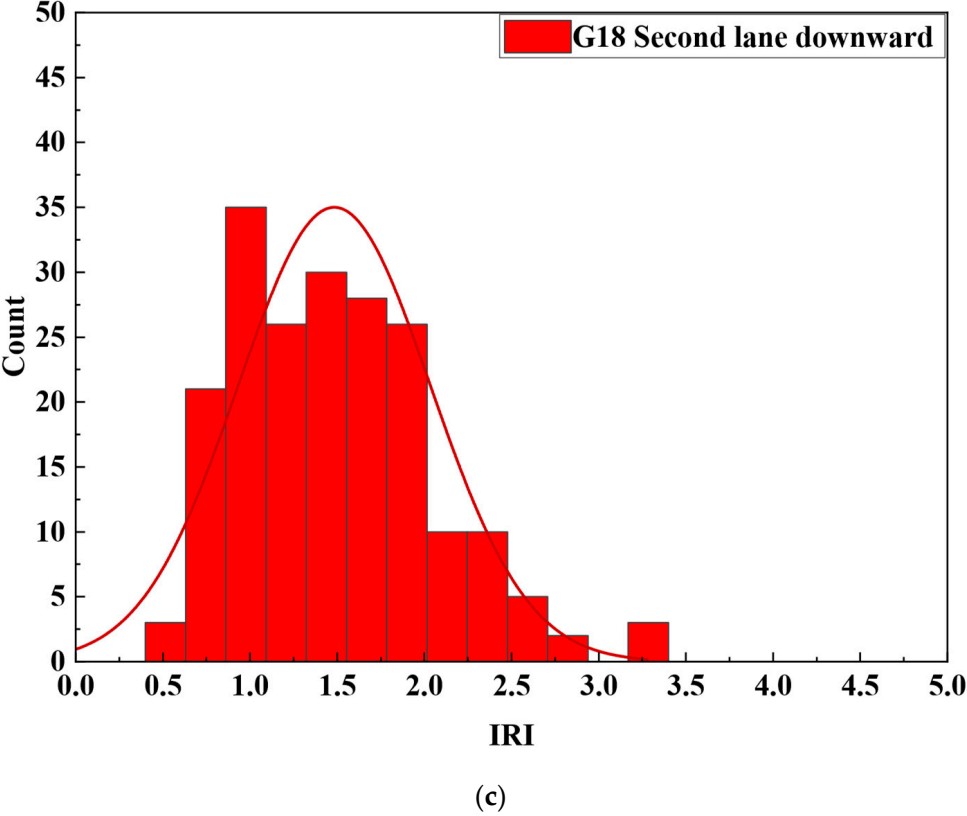

(**c**)

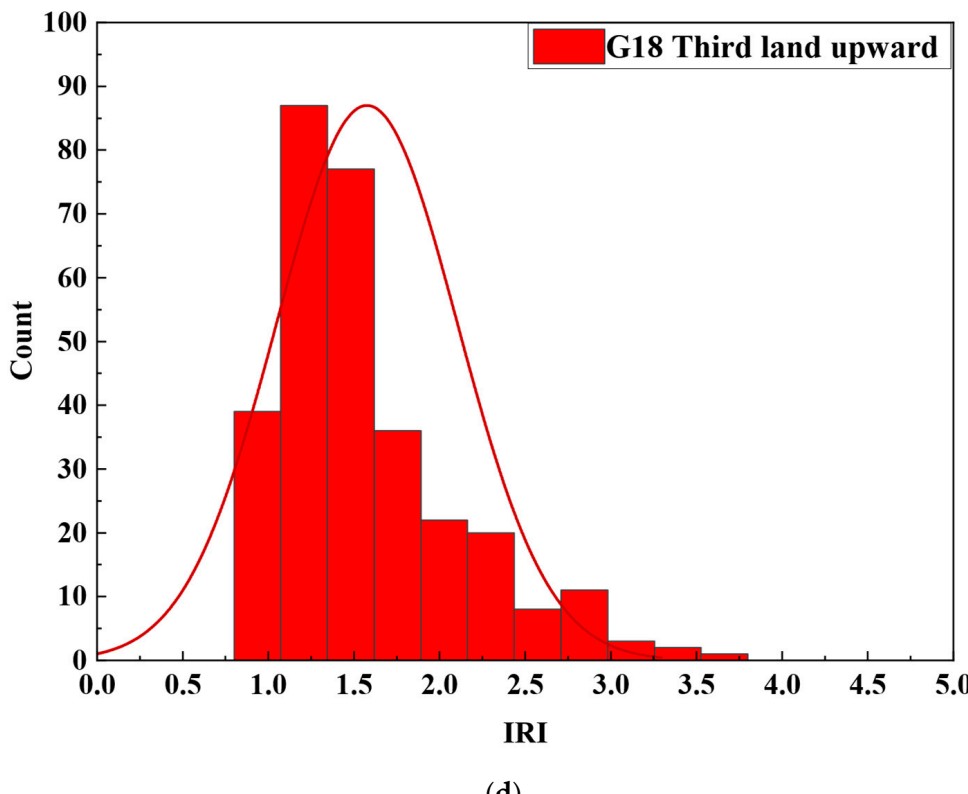

(**d**)

**Figure 4.** *Cont.*

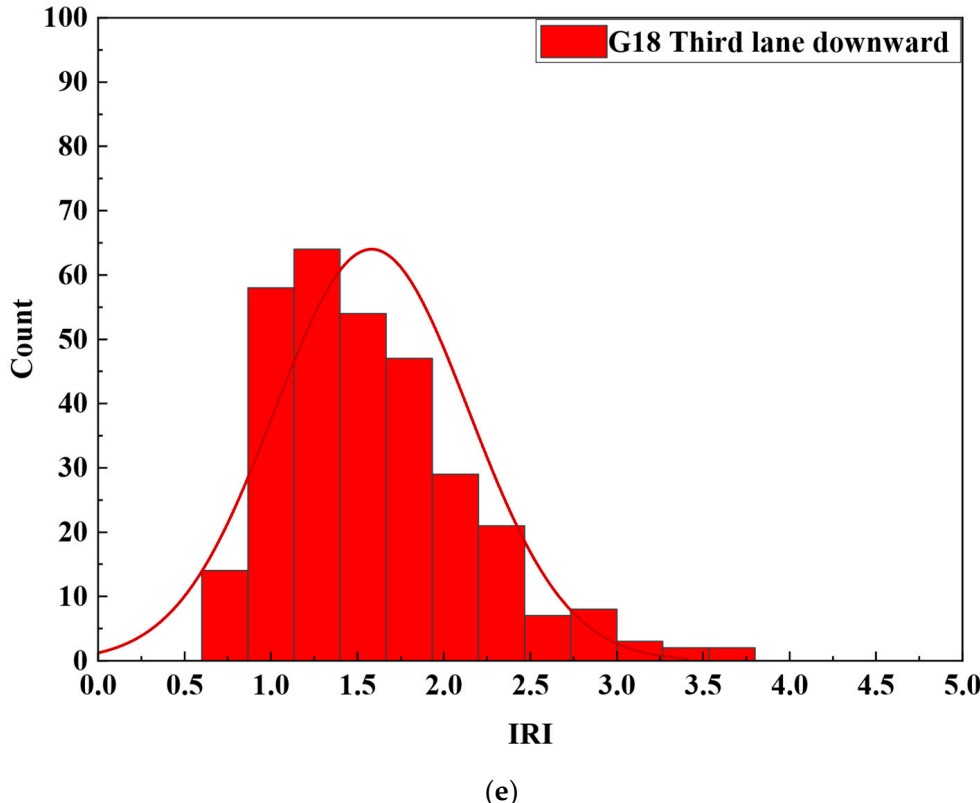

(**e**)

**Figure 4.** Data distribution of the LTPP database and G18 database. (**a**) LTPP; (**b**) G18 second lane upward; (**c**) G18 second lane downward; (**d**) G18 third lane upward; (**e**) G18 third lane downward.

The sections used in this study are both asphalt concrete pavement. Six variables, including road age, annual average daily traffic volume, annual average temperature, annual total rainfall, asphalt surface thickness, and road base thickness, are chosen as input features in the model, with IRI as the output feature. These selected features are all important variables that highly affect road deterioration. Tables 1 and 2 summarize detailed statistical data for the LTPP and G18 datasets.

**Table 1.** Statistical data of LTPP features.

|  | Road Age (Year) | AADT (Vehicle/Day) | Temperature (°C) | Precipitation (mm) | Base Thickness (mm) | Asphalt Layer Thickness (mm) | IRI (m/km) |
|---|---|---|---|---|---|---|---|
| Mean | 19.78 | 5330.41 | 12.23 | 724.31 | 504.99 | 176.59 | 1.18 |
| Std | 12.31 | 6353.96 | 5.22 | 443.26 | 497.50 | 72.80 | 0.49 |
| Min | 1 | 63.00 | 0.90 | 10.50 | 25.40 | 12.70 | 0.33 |
| 25% quartile | 8 | 1151.00 | 8.00 | 320.39 | 304.80 | 119.40 | 0.85 |
| 50% quartile | 20 | 2829.80 | 10.90 | 696.70 | 416.60 | 162.50 | 1.07 |
| 75% quartile | 30 | 6967.74 | 14.50 | 1052.80 | 604.50 | 213.40 | 1.35 |
| Max | 52 | 45,909.00 | 25.70 | 2447.69 | 2456.00 | 444.60 | 4.11 |

**Table 2.** Statistical data of G18 features.

|  | Road Age (Year) | AADT (Vehicle/Day) | Temperature (°C) | Precipitation (mm) | Base Thickness (mm) | Asphalt Layer Thickness (mm) | IRI (m/km) |
|---|---|---|---|---|---|---|---|
| Mean | 1.92 | 21,105.38 | 12.97 | 547.58 | 540 | 180 | 1.16 |
| Std | 0.79 | 1736.77 | 0.28 | 167.63 | 0 | 0 | 0.48 |
| Min | 1 | 19,154.37 | 12.54 | 422.66 | 540 | 180 | 0.39 |

**Table 2.** *Cont.*

|  | Road Age (Year) | AADT (Vehicle/Day) | Temperature (°C) | Precipitation (mm) | Base Thickness (mm) | Asphalt Layer Thickness (mm) | IRI (m/km) |
|---|---|---|---|---|---|---|---|
| 25% quartile | 1 | 19,286.29 | 12.75 | 422.66 | 540 | 180 | 0.84 |
| 50% quartile | 2 | 22,723.40 | 12.87 | 513.59 | 540 | 180 | 1.05 |
| 75% quartile | 2 | 22,723.40 | 13.17 | 513.59 | 540 | 180 | 1.34 |
| Max | 3 | 22,804.41 | 13.42 | 880.11 | 540 | 180 | 4.79 |

## 4. Methodology

Transfer learning aims to transfer data from the source domain to the target domain with a certain correlation. As shown in Figure 5, although the data distribution between the source domain and the target domain is different, transfer learning can help machine learning models transfer learned knowledge from the source domain to the target domain and assist the target domain training model in improving the accuracy of the target domain model [43].

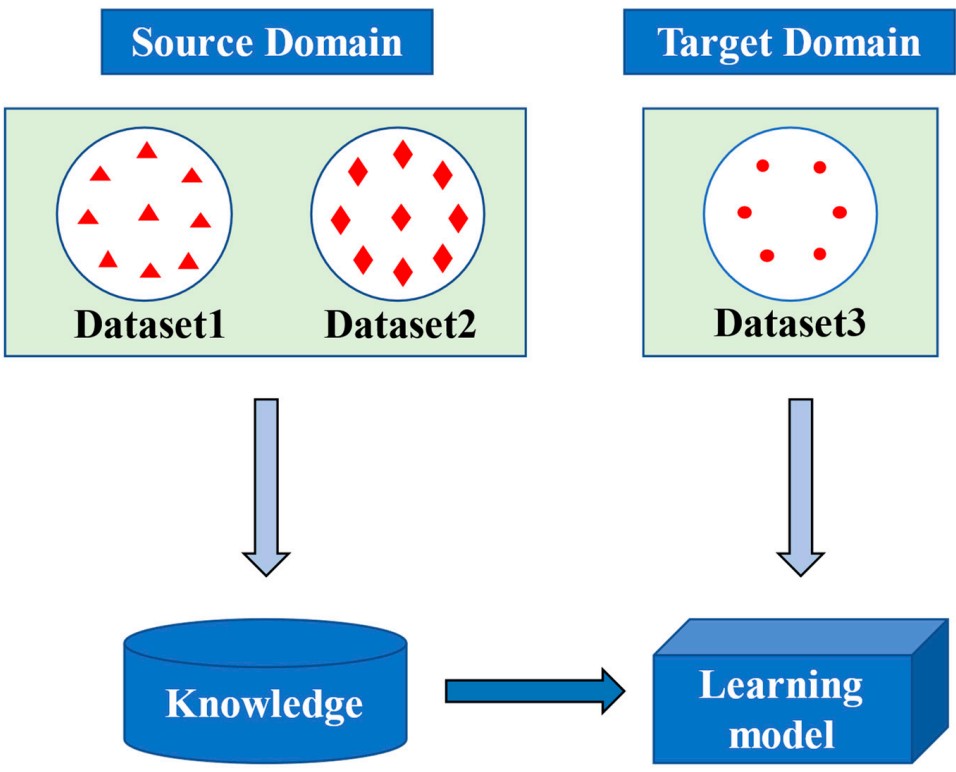

**Figure 5.** Learning process of transfer learning.

### 4.1. AdaBoost.$R^2$

AdaBoost is a machine learning algorithm based on boosting. Its main task is to generate a strong classifier from a series of weak classifiers to improve prediction accuracy [44]. The AdaBoost mechanism involves training the weak learner from training samples with the current weight and updating the weight of the training samples based on the error rate. The weights of misclassified samples from the previous weak learner are increased, while the weights of correctly classified samples are decreased and used again to train the next weak learner. In addition, a new weak learner is added in each round of iteration, and the weak learner will not be integrated into the final strong learner until a predetermined error rate small enough or a prespecified maximum number of iterations has been attained. Drucker [45] compared boosting with a single learner and bagging

on the Friedman function, showing that boosting has better performance in dealing with regression problems.

### 4.2. TrAdaBoost.$R^2$

As an improvement of AdaBoost.$R^2$, TrAdaBoost.$R^2$ is an instance-based transfer learning algorithm. Dai et al. [46] extended AdaBoost to the field of transfer learning. In TrAdaBoost.$R^2$, the training set is mainly divided into two groups, including source domain data and target domain data. It allows the source domain to participate in training and extract knowledge to help the target domain build models. Although the distributions of the assumed source and target domain data are different, the algorithm automatically adjusts the weights of the source domain data. The source domain data with a similar distribution as the target domain data will be given higher weights, which will improve the accuracy of the model. Figure 6 shows the conceptual diagram of the TrAdaBoost.$R^2$ algorithm, blue circles represent source domain data red triangles represent target domain data. Dai et al. [46] investigated different ratios between training data of the same distribution and different distributions as variables. The results show that the error rate of TrAdaBoost is consistently lower than other support vector machine (SVM)-based algorithms in different datasets.

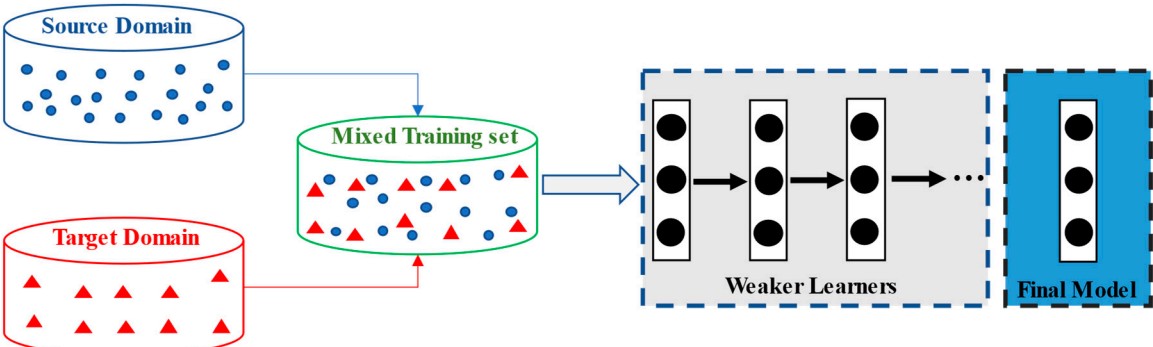

**Figure 6.** Conceptual diagram of the TradaBoost.$R^2$ algorithm.

The main difference between AdaBoost and TrAdaBoost is the strategy for updating the weights of training data samples. For the target domain, both algorithms reduce the weight of high error rate instances and increase the weight of low error rate instances. However, for the source domain, TradaBoost.$R^2$ reduces the weight of high error instances because they may conflict with test data and increases the weight of low error instances to contribute more to the learning process.

### 4.3. Two-Stage TrAdaBoost.$R^2$

Since the original TrAdaBoost algorithm is a classification algorithm, the TrAdaBoost framework is combined with AdaBoost.$R^2$, which becomes TrAdaBoost.$R^2$, to predict regression problems. However, Pardoe and Stone [47] found two problems when implementing the TrAdaBoost.$R^2$ algorithm. Firstly, the weight of source instances similar to the target data is often reduced to zero during the process of transfer learning. Secondly, the target instance that is least similar to the source data is given a higher weight during the training process. Therefore, Pardoe and Stone [47] introduced the two-stage TrAdaBoost.$R^2$ to deal with the two main problems of TrAdaBoost. To avoid the problem of overfitting, the weight of the source instance in the first stage of the Two-stage TrAdaBoost.$R^2$ algorithm is adjusted to a point specified by cross-validation through N steps. In the last iteration step, the total weight is reduced to zero. In the second stage, the weight of the source instance remains unchanged while updating the weight of the target instance by implementing normal AdaBoost. The main steps of the algorithm are as follows:

a.   Initialize weights and assign weight distribution $D_1$ to the training dataset, setting the initial weight vector $\omega_i^1$ as:

$$D_1 = \left(\omega_1^1, \ldots, \omega_{s+t}^1\right); \; \omega_i^1 = \frac{1}{s+t} \; for 1 \leq i \leq s+t \tag{1}$$

where $D_1$ is the weight distribution of the training data, $\omega_i^1$ is the weight distribution of each data weight vector under $D_1$, $s$ is the dataset T$_{source}$ (of size s), $t$ is the dataset T$_{target}$ (of size t). For $n$ = 1, 2, 3, ..., N:

b.   Call a learner $G_n(x)$ from the training dataset T with the weight distribution $D_n$.

c.   Call AdaBoost.R$^2$ with T = T$_{source}$ + T$_{target}$, a base regression estimator G(x), and the weight vector $\omega_j$. Tsource stays unchanged. Calculate the error$_j$ of Model$_j$ using F-fold cross-validation.

d.   Call a learner $G(x)$ with T with the weight distribution $D_j$.

e.   Calculate the adjusted error $e_i^j$ of each instance in T using AdaBoost.R$^2$.

f.   Update the weight vector and the weight distribution.

$$\omega_i^{j+1} = \begin{cases} \dfrac{\omega_i^j \beta_j^{e_i^j}}{Z_j} & 1 \leq i \leq s, \; for \; \text{Tsource} \\[2mm] \dfrac{\omega_i^j}{Z_j} & s+1 \leq i \leq s+t, \; for \; \text{Ttarget} \end{cases} \tag{2}$$

where $Z_j$ is the normalizing constant, $\beta_j$ is the weighting factor to result in a certain total weight for the target instances, selected such that the observed weight of Ttarget is $\frac{t}{s+t} + \frac{j}{N-1}\left(1 - \frac{t}{s+t}\right)$.

g.   Determine the output of the resulting Model$_j$:

$$f(x) = \text{Model}_j = f^j(x), \text{ where } j = \text{argmin}_i \; error_i$$

### 4.4. Decision Tree

A Decision Tree is a supervised learning method that summarizes decision rules from a series of data with features and labels and presents these rules with the structure of a tree graph to solve classification and regression problems [48–50]. The decision tree is mainly composed of one root node, multiple decision nodes, and leaf nodes. The root node is the starting node, which includes all samples. The commonly used decision trees are iterative dichotomiser 3(ID3) [51,52], classification and regression trees based on information gain ratio (C4.5) [53,54], and Classification and Regression Tree (CART) [55–58]. The CART algorithm uses the division criteria of the Gini coefficient to deal with continuous values in the regression problem. However, practice has proven that the fully grown regression tree is prone to overfitting, and the prediction effect on new samples is poor. Therefore, it is necessary to reduce the fully grown regression tree using pruning criteria to obtain the optimal regression tree model. In addition, CART is suitable for large-scale datasets, especially with complex samples and more variables. Since CART can solve both classification and regression problems, it is used as a weak classifier for IRI prediction in the above ensemble learning model.

### 4.5. Particle Swarm Optimization (PSO) Algorithm

Particle swarm optimization (PSO) [59] is an intelligent algorithm that simulates the predation characteristics of birds. It stimulates the process of birds looking for food in a certain area. Particle swarm optimization solves global optimization problems by simulating biological populations [60]. In the PSO algorithm, each individual in the group is regarded as a particle, and each particle moves at a certain speed in a given search space. The algorithm can update speed and iteration dynamically according to the movement of the particles themselves and the surrounding particles. The specific process is shown

in Figure 7. The performance of the model depends largely on the hyperparameters in the model training process. Kawther et al. [61] applied the PSO algorithm to optimize the maintenance schemes. The results show that the PSO optimization scheme can improve road conditions and save costs. Li et al. [1] used the Gated Recurrent Unit (GRU) neural network enhanced by the particle swarm optimization (PSO) algorithm to predict the pavement performance parameters based on seven years of historical data from a highway in China. The results show that the predicted pavement performance parameters are significantly consistent with the annual measurement data. Therefore, it is necessary to adjust the hyperparameters of the model, such as decision tree depth, learning rate, number of estimators, number of iterations, etc. The purpose of model hyperparameter tuning is to obtain a set of optimal parameter values that produce the best predictive performance. This process can greatly reduce human influence during model training and automatically find the best hyperparameters.

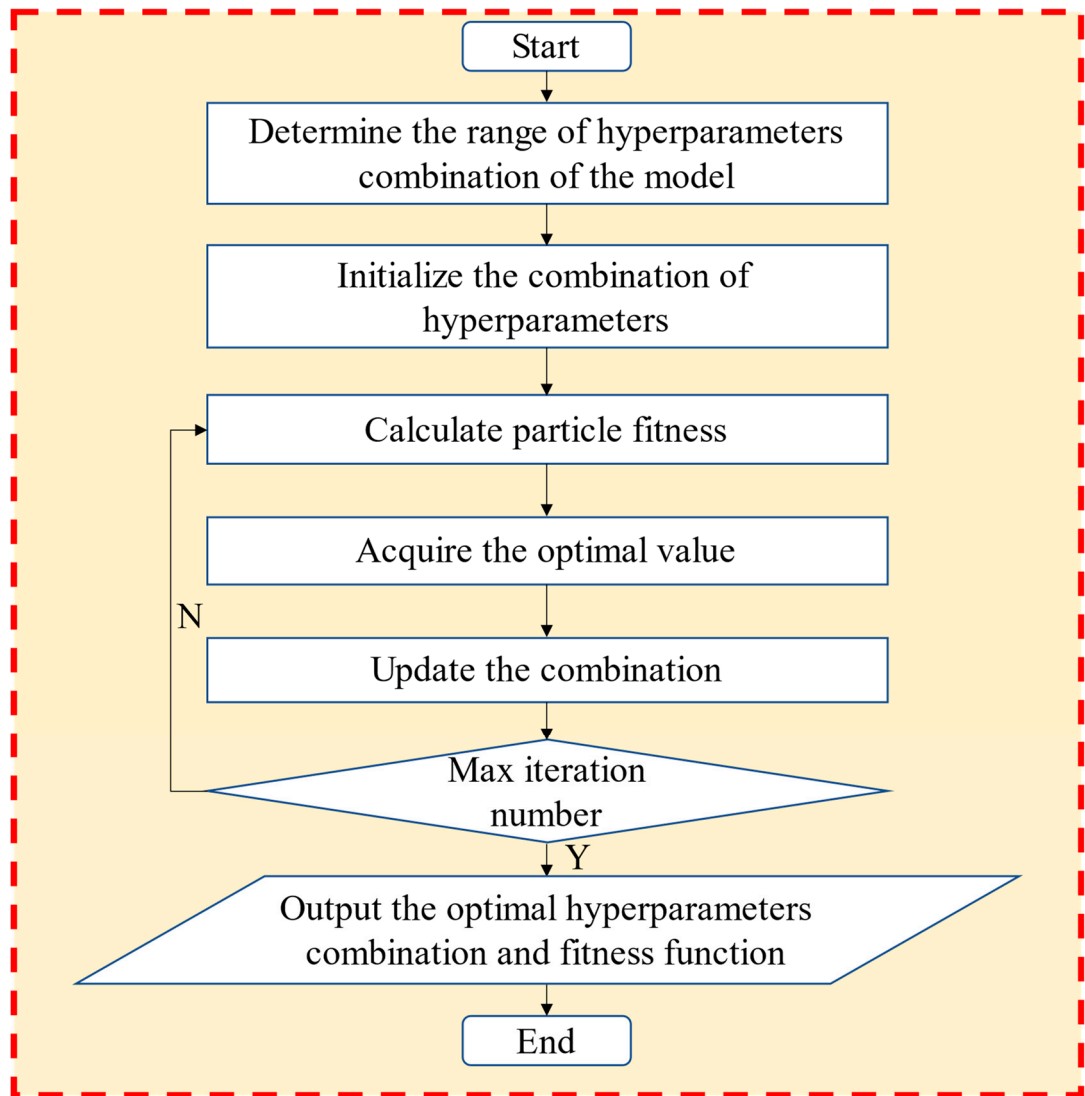

**Figure 7.** Process of finding optimal hyperparameters using the particle swarm optimization algorithm.

### 4.6. Input and Output Variables

The deterioration of pavement performance results from the interaction of environmental factors and vehicle loads. The influence of environmental factors, vehicle load, and road structure should be considered in the prediction model. In this study, annual average temperature, annual rainfall, annual average daily traffic, road age, asphalt layer thickness,

road base thickness, and the previous year's road performance index are all used as the pavement performance prediction model's input variables.

Pavement performance data are time series data as performance decreases over time. When the pavement performance model is established, the step size of time is 1, which means that the pavement performance index in the second year is predicted by the predetermined data froom the previous year:

$$I_t = f(C_{t-1}, I_{t-1}) \tag{3}$$

where *I* is the road performance index (IRI), *C* is the feature values, which include annual average temperature, annual rainfall, road age, annual average daily traffic volume (AADT), asphalt layer thickness, road base thickness, and t is the year.

### 4.7. Data Preprocessing

In this study, 2611 sets of data were chosen as the source domain from the LTPP database. A total of 1016 sets of data were chosen as target domains, with 202 for the second lane upward, 199 for the second lane downward, 306 for the third lane upward, and 309 for the third lane downward. Each set contains two consecutive years of data with all variables. To simulate transfer to a newly built road, the first two years' data, thus 2010 and 2011, are chosen as the training set, and the data from 2012 are chosen as the validation set. The data of the same milestone number in the training set are arranged vertically, as shown in Figure 8, in which *C* represents the environment, load, and other variables, and I represents the pavement performance index value. The pavement needs to be maintained during service, and this may lead to abnormal fluctuations in the pavement performance index. Some other abnormal fluctuations may be caused by human factors. Both abnormal fluctuations caused by maintenance work and human factors are removed in the preprocessing stage.

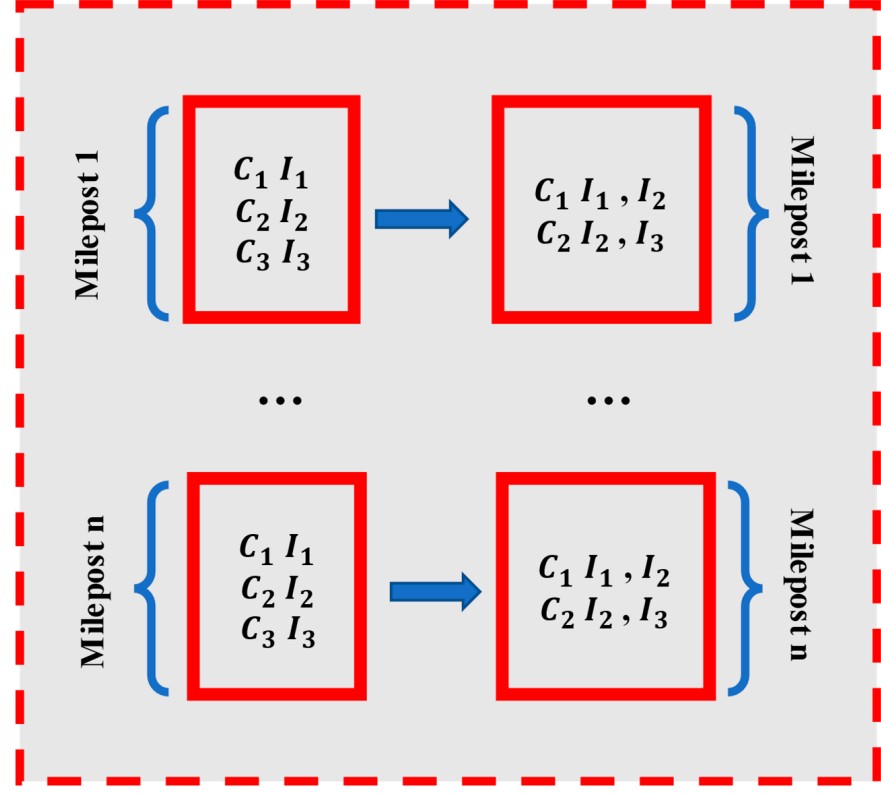

**Figure 8.** Data preprocessing.

## 5. Results

### 5.1. Model Evaluation Indexes

The Root Mean Square Error (RMSE), the Coefficient of Determination ($R^2$), and the Mean Absolute Percentage Error (MAPE) are used as evaluation indexes for the model.

$$\text{RMSE} = \sqrt{\frac{1}{n}\sum_{i=1}^{n}(Y_i - y_i)^2} \tag{4}$$

$$\text{MAPE} = \frac{100\%}{n}\sum_{i=1}^{n}\left|\frac{Y_i - y_i}{y_i}\right| \tag{5}$$

$$R^2 = 1 - \frac{\sum(y_i - Y_i)^2}{\sum(\overline{Y_i} - y_i)^2} \tag{6}$$

where $n$ is the amount of data in the test set, $y_i$ is a single true value of the test set, $Y_i$ is a single predicted value of the test set, and $\overline{Y_i}$ is the sample expectation of true value.

### 5.2. Prediction Result

In this study, three models: the decision tree model, AdaBoost.$R^2$, and the Two-stage TrAdaBoost.$R^2$ model are compared to predict the IRI of the G18 highway based on LTPP data as the source domain. To compare the improvement of the training dataset, the prediction results using only G18 data are also compared, as shown in Table 3. It is shown from Table 3 that the Two-stage TrAdaBoost.$R^2$ model generally shows better prediction performance than Adaboost.$R^2$ and decision tree regression models. The RMSE and MAPE of the Two-stage TrAdaBoost.$R^2$ model are smaller than those of AdaBoost.$R^2$, the regression decision tree model, and the local regression decision tree model. The $R^2$ of the Two-stage TrAdaBoost.$R^2$ model is higher than that of the other three models. The AdaBoost $R^2$ model has the second-best performance of all models. The average $R^2$ of the Two-stage TrAdaBoost.$R^2$ model for the four lanes is 0.76, which is 7.29% higher than the average $R^2$ of the AdaBoost.$R^2$ model, 21.2% higher than the average $R^2$ of the regression decision tree model, and 36.2% higher than the local regression decision tree model. These results show that the Two-stage TrAdaBoost.$R^2$ model has the best performance in predicting the IRI. By comparing the decision tree model using different training datasets, it is shown that the model trained with LTPP data together with local data has higher accuracy compared with the model trained only with local data.

**Table 3.** Prediction results.

| | Two-Stage TrAdaBoost.$R^2$ | | | AdaBoost.$R^2$ | | | Decision Tree | | | Decision Tree (Local) | | |
|---|---|---|---|---|---|---|---|---|---|---|---|---|
| | **RMSE** | **MAPE** | **$R^2$** | **RMSE** | **MAPE** | **$R^2$** | **RMSE** | **MAPE** | **$R^2$** | **RMSE** | **MAPE** | **$R^2$** |
| Second lane upward | 0.227 | 0.1107 | 0.83 | 0.261 | 0.1810 | 0.78 | 0.300 | 0.1551 | 0.71 | 0.316 | 0.1768 | 0.67 |
| Second lane downward | 0.273 | 0.1208 | 0.75 | 0.321 | 0.1449 | 0.65 | 0.347 | 0.164 | 0.60 | 0.389 | 0.1862 | 0.50 |
| Third lane upward | 0.246 | 0.1001 | 0.78 | 0.297 | 0.1286 | 0.68 | 0.340 | 0.1396 | 0.58 | 0.360 | 0.1480 | 0.53 |
| Third lane downward | 0.323 | 0.1037 | 0.67 | 0.344 | 0.1107 | 0.62 | 0.358 | 0.1241 | 0.59 | 0.385 | 0.1465 | 0.52 |

Figure 9a,c,e,g shows scatter plots of the predicted and measured IRI for the four lanes. The horizontal axis and vertical axis represent the measured actual values and the predicted values, and the 1:1 line is plotted as the reference line. It is shown from the figure that the scatter points of the Two-stage TrAdaBoost.$R^2$ model are evenly distributed around the 11 line, and most of the scatter points range between 0.5 and 3.5, which means that the Two-stage TrAdaBoost.$R^2$ model matches the actual data very well. Compared with the Two-stage TrAdaBoost.$R^2$ model, the scattering of the other three models is more scattered around the 1:1 line and more dispersed above or below the reference line, indicating that the prediction accuracy of the other three models is lower than that of the

Two-stage TrAdaBoost.$R^2$ model. It also can be seen that the predicted values for the second lane are distributed closer to the reference line compared with the third lane for both directions, which indicates that the model has different accuracy levels for different lanes. For a more intuitive comparison of the distribution of the predicted data, Figure 9b,d,f,h plots box plots of the true values and the predicted results of the four models to compare the distribution characteristics of the data, where black dots represent data outliers. The quality of the model can be assessed through statistical analysis comparing observed and predicted values. First, from the box plots, it can be seen that the 25–75% data distribution of the predicted values for the Two-stage TrAdaBoost.$R^2$ model is more concentrated, and the boxes of the Two-stage TrAdaBoost.$R^2$ positions are the same. The other three models for the four lanes are all notably lower than the predicted values of the Two-stage TrAdaBoost.$R^2$ model. In addition, the number of outliers in the predicted values for the Two-stage TrAdaBoost.$R^2$ model is fewer than that of the other four models. This shows that the Two-stage TrAdaBoost.$R^2$ model can effectively transfer data and predict pavement performance.

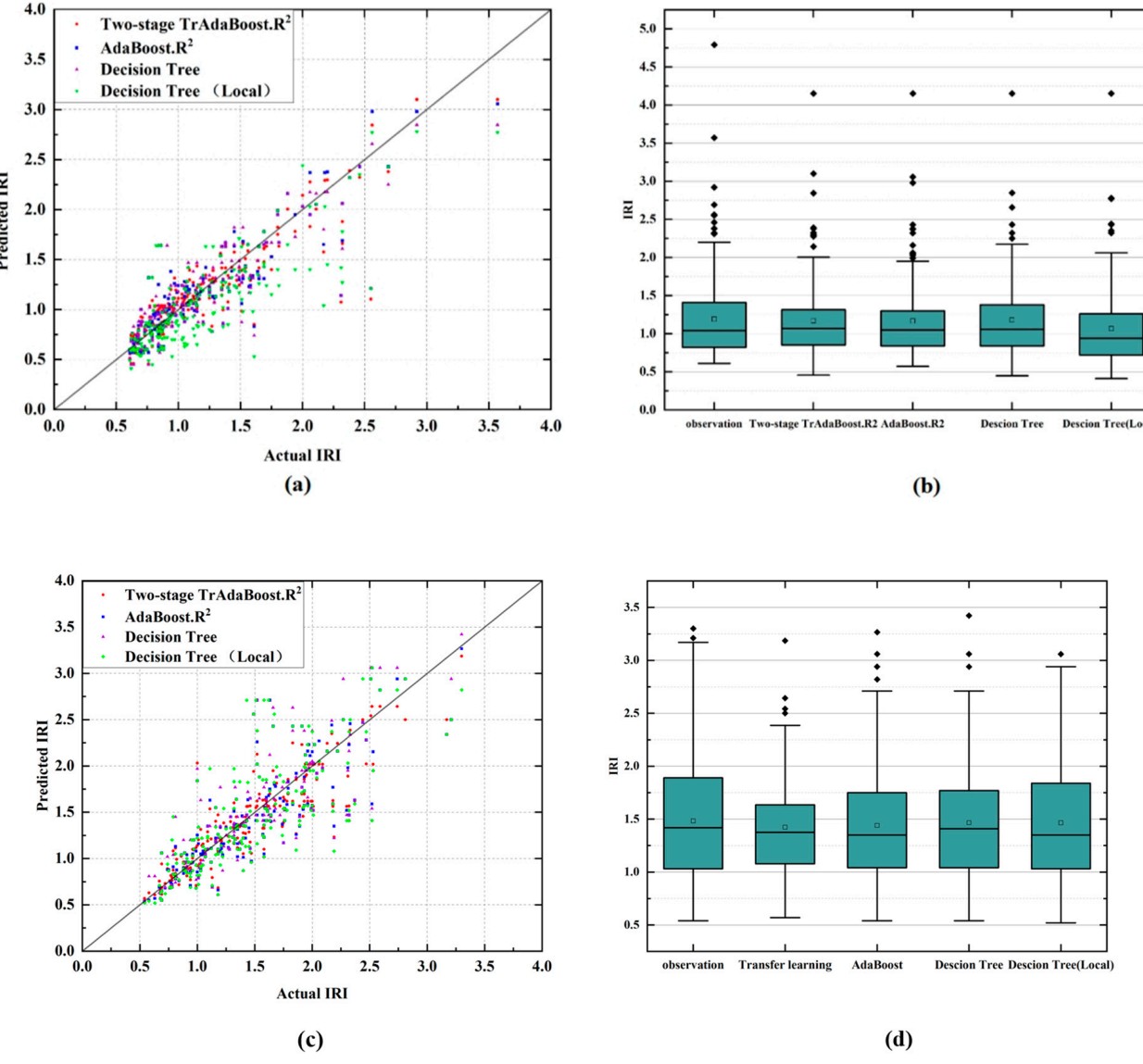

**Figure 9.** *Cont.*

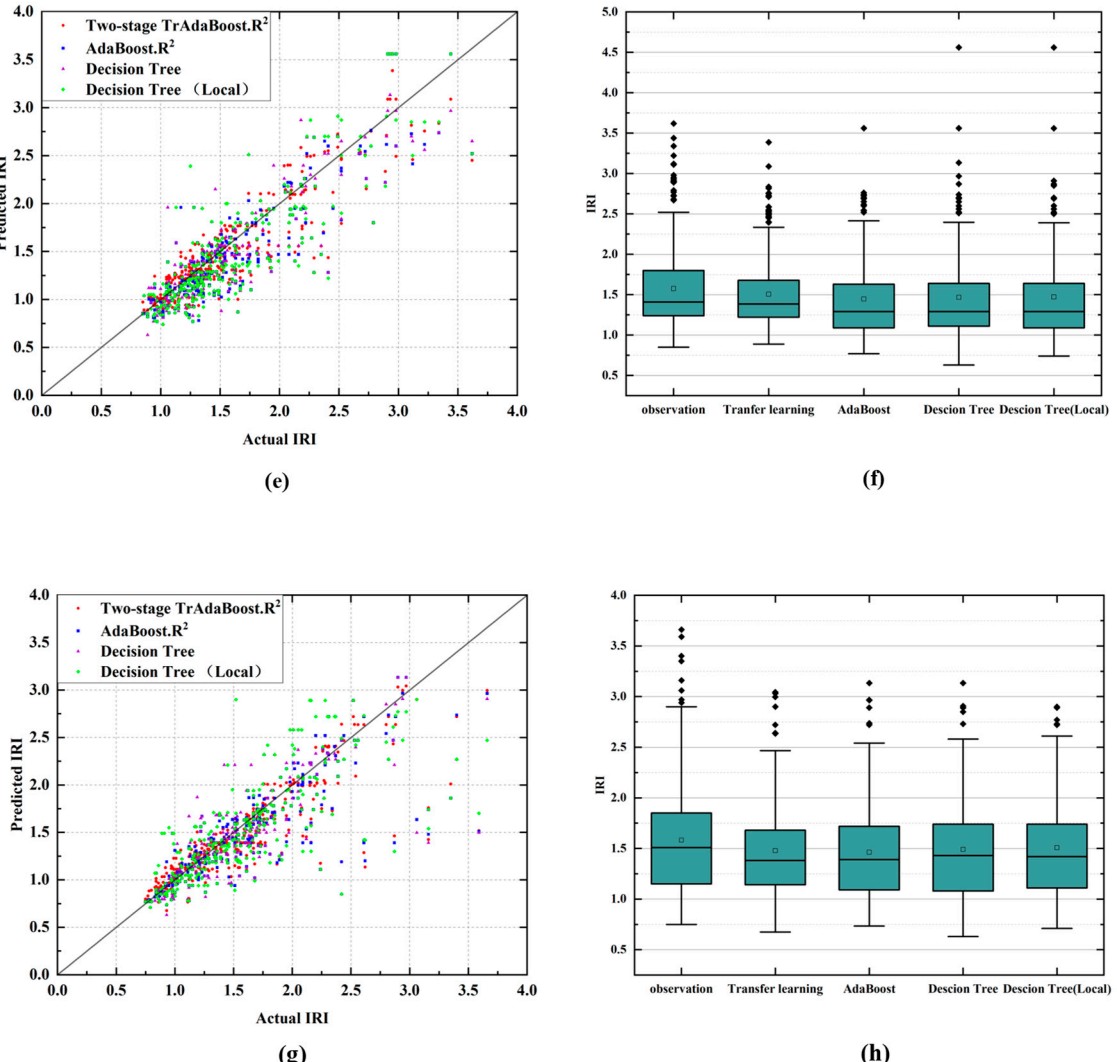

**Figure 9.** Prediction results of IRI. (**a**) The scattergram of predicted IRI and actual IRI of the second lane upward; (**b**) the box plot of predicted IRI and actual IRI of the second lane upward; (**c**) the scattergram of predicted IRI and actual IRI of the second lane downward; (**d**) the box plot of predicted IRI and actual IRI of the second lane downward; (**e**) the scattergram of predicted IRI and actual IRI of the third lane upward; (**f**) the box plot of predicted IRI and actual IRI of the third lane upward; (**g**) the scattergram of predicted IRI and actual IRI of the third lane downward; (**h**) the box plot of predicted IRI and actual IRI of the third lane downward.

Figure 10, shows the Taylor diagrams of the four models. The purple lines represent the Pearson correlation coefficient, the green lines represent the RMSE error, and the black lines represent the standard deviation. The standard deviation and RMSE of different indicators are scaled from 0 to 1, according to the proportion, to compare the four models at the same time. In the Taylor diagram, the degree of agreement between the observed and predicted IRI is revealed by the evaluation indicators of the four models. The figure represents three index values of the predicted results. In the Taylor diagram, each model is represented by a small circle, and a model closer to the observed value has better quality and higher prediction accuracy. It can be seen from Figure 10 that the predicted values of the Two-stage TrAdaBoost.$R^2$ model for both lanes are more accurate than those of the other models.

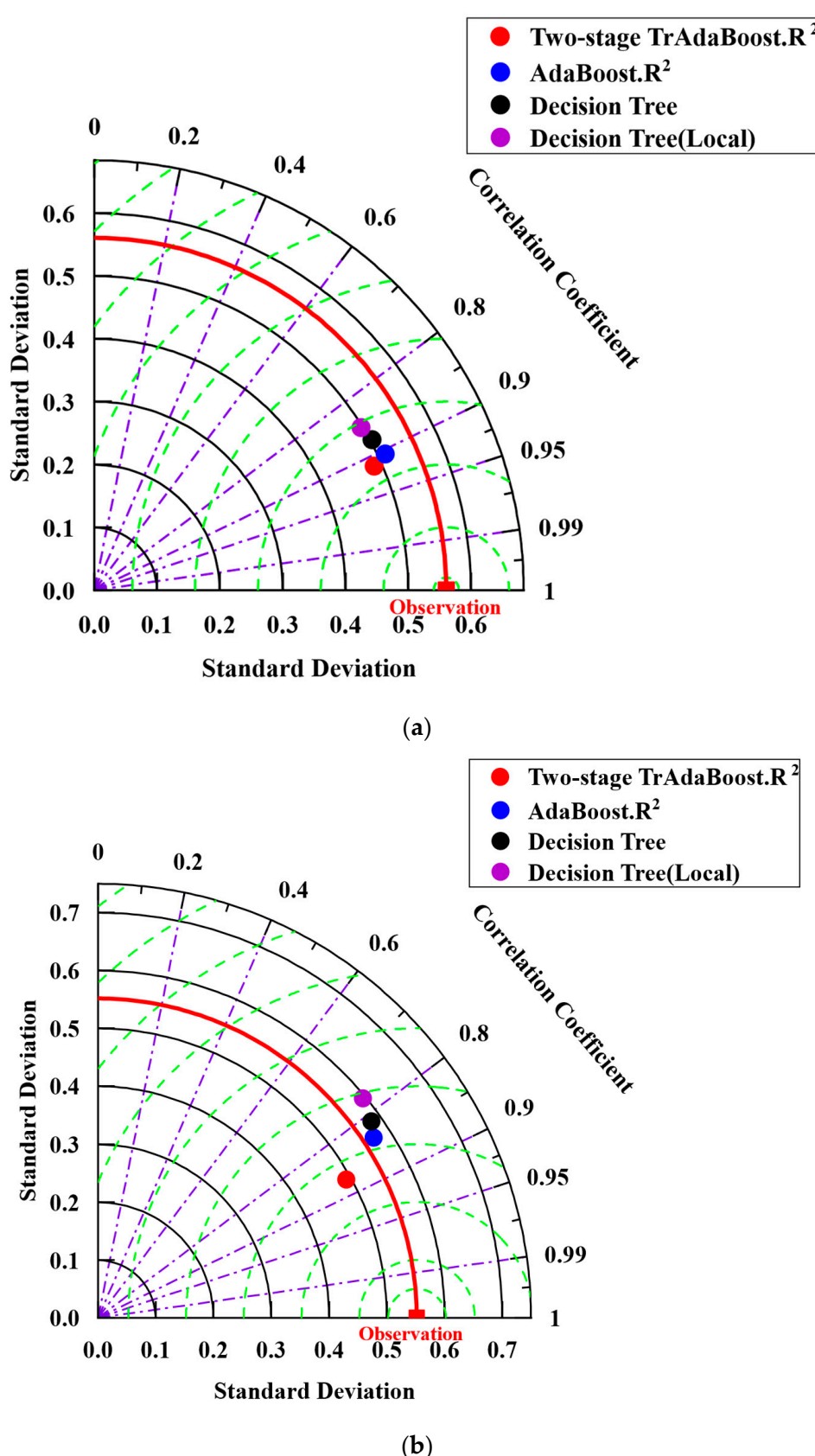

**Figure 10.** *Cont.*

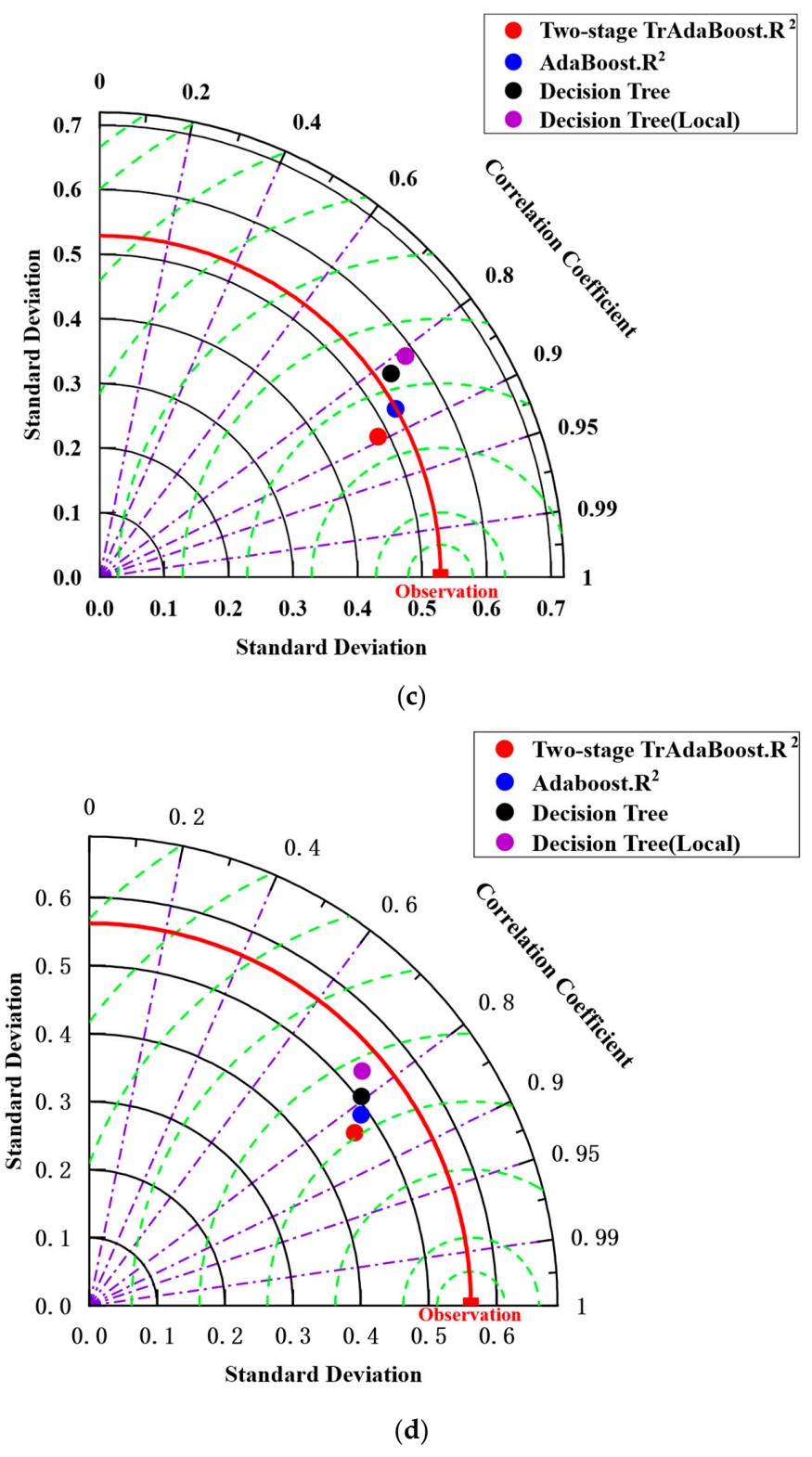

**Figure 10.** The Taylor diagrams of the four models. (**a**) G18 second lane upward; (**b**) G18 second lane downward; (**c**) G18 third lane upward; (**d**) G18 third lane downward.

### 5.3. Impact of Training Dataset

To study the impact of source and target domain training datasets on the transfer learning model, a series of experiments were conducted. These experiments obtained different results for the Two-stage TrAdaBoost.$R^2$ model by gradually reducing the size of the source and target domain datasets. Specifically, the experiments included 100%, 75%,

50%, and 25% of the data in the source domain, using all the data in the target domain. The model was trained using 100%, 75%, 50%, and 25% of the target domain data and all the source domain data, as shown in Figure 11. Figure 12 shows the prediction results for the four lanes, and specific statistical analysis results are shown in Tables 4 and 5.

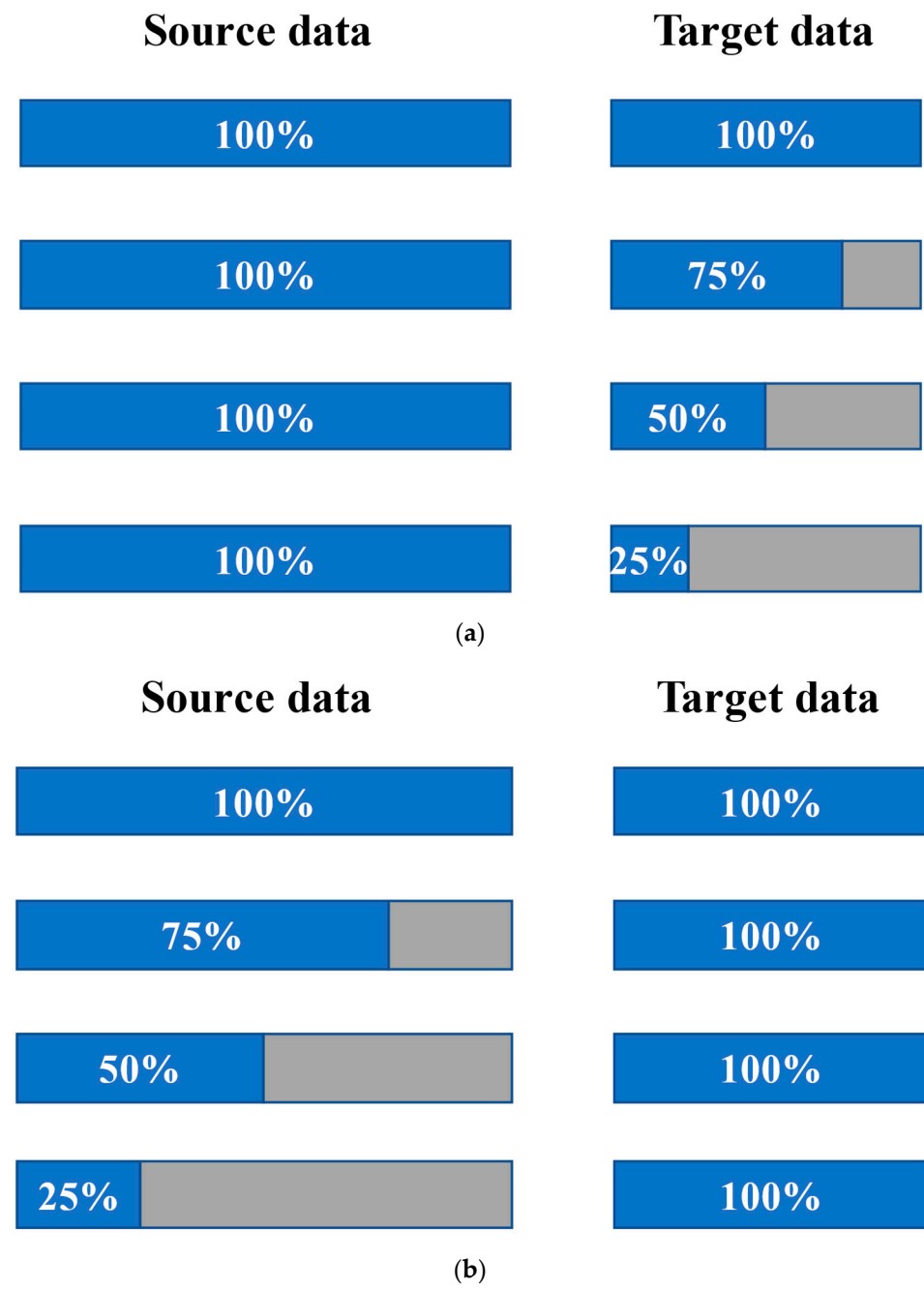

**Figure 11.** Dataset change diagram. (**a**) Target dataset variable; (**b**) source dataset variable.

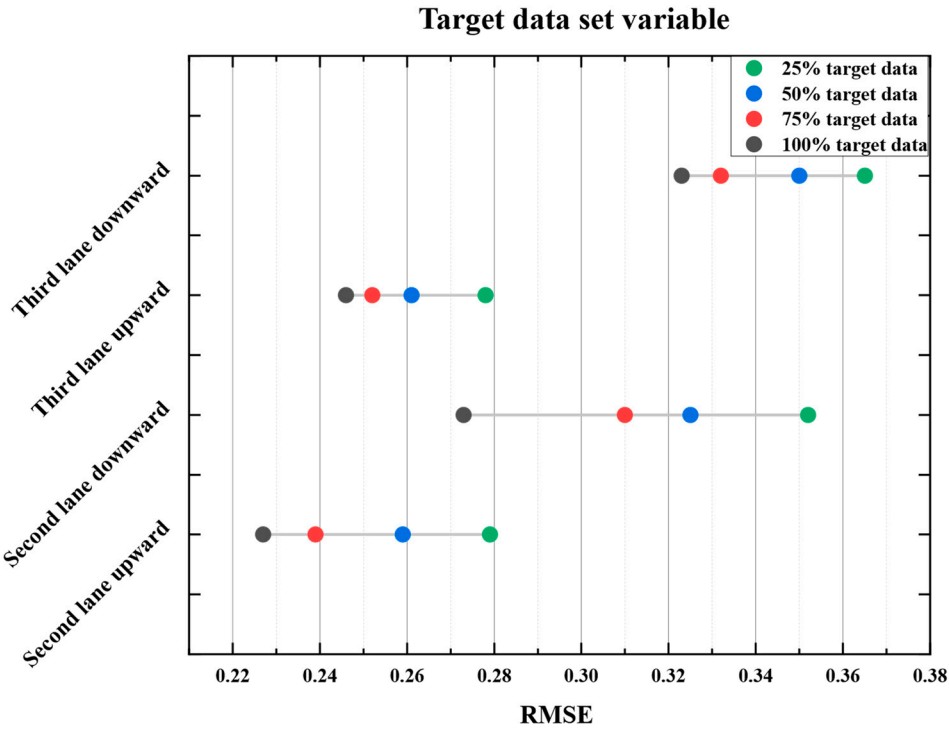

(**a**)

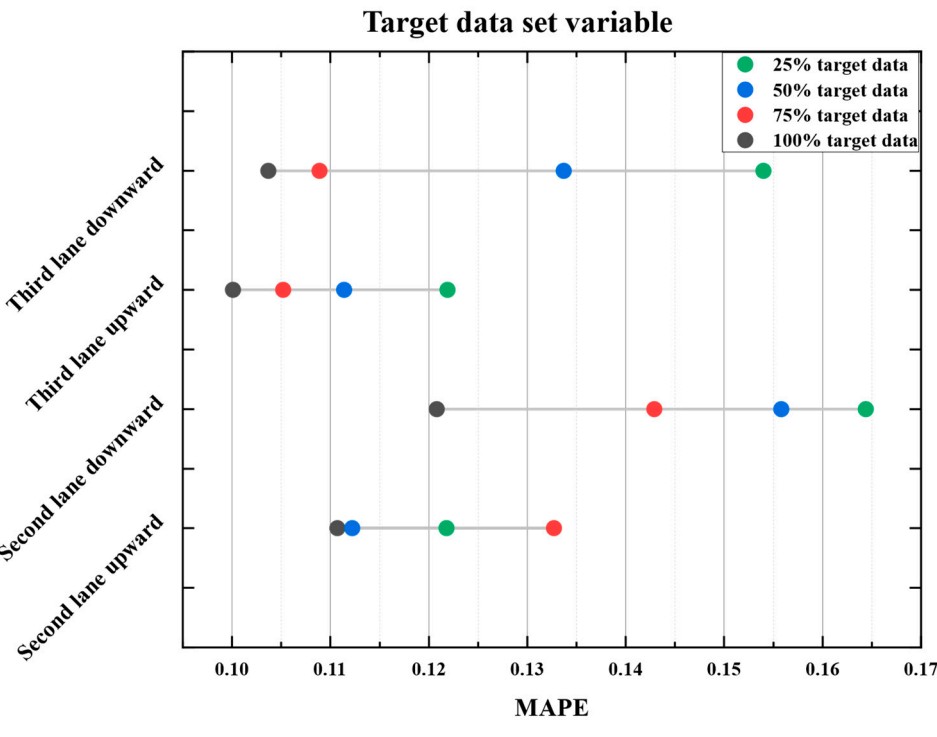

(**b**)

**Figure 12.** *Cont.*

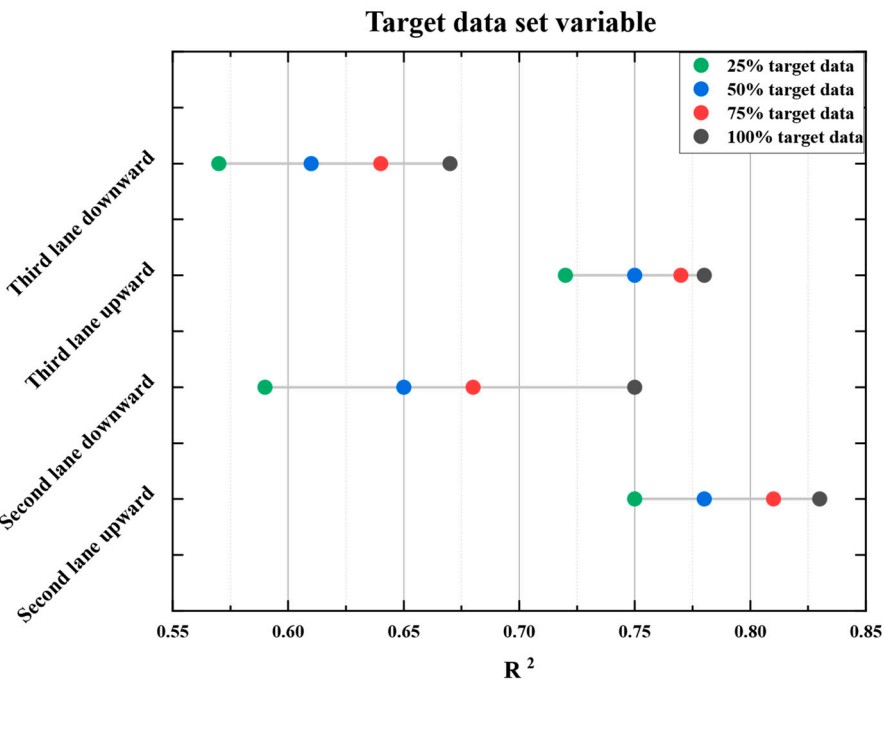

(**c**)

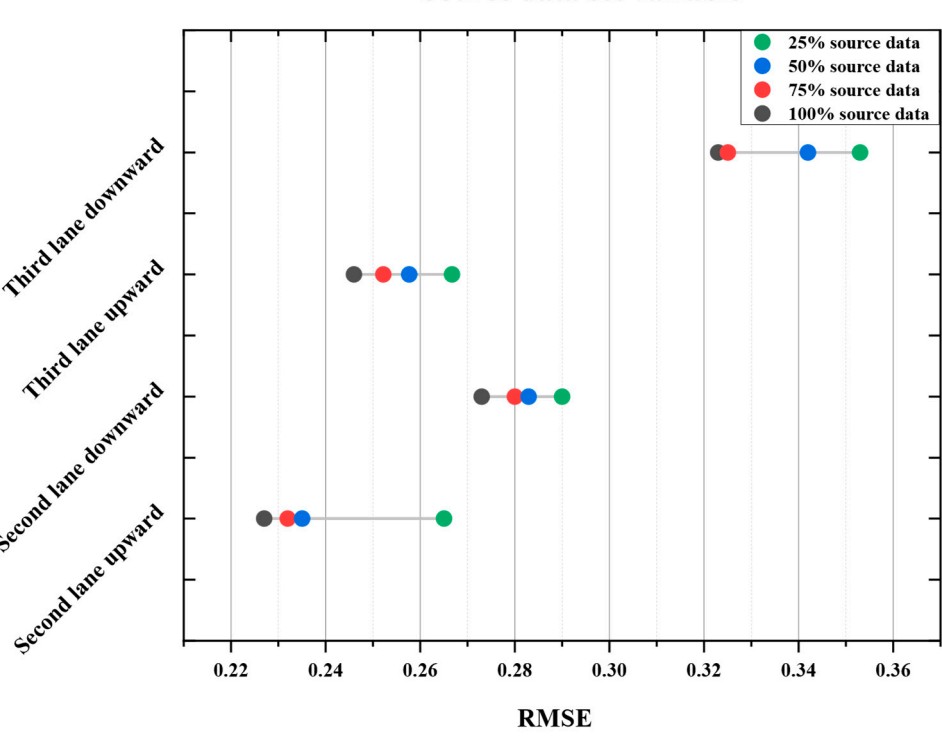

(**d**)

**Figure 12.** *Cont.*

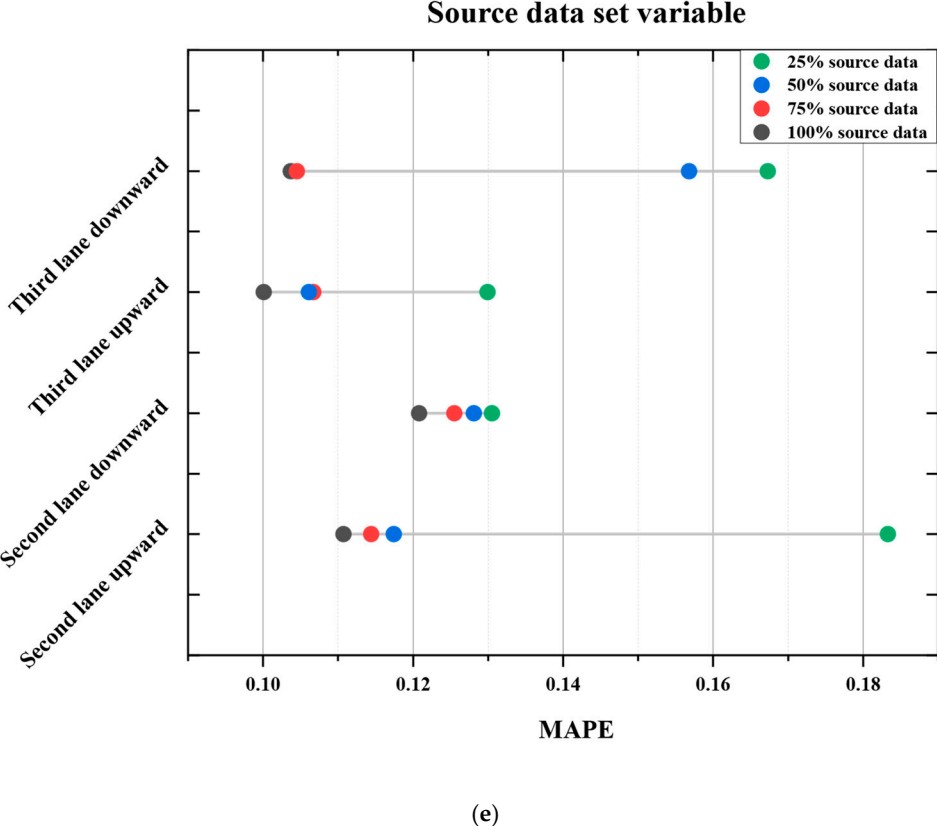

(**e**)

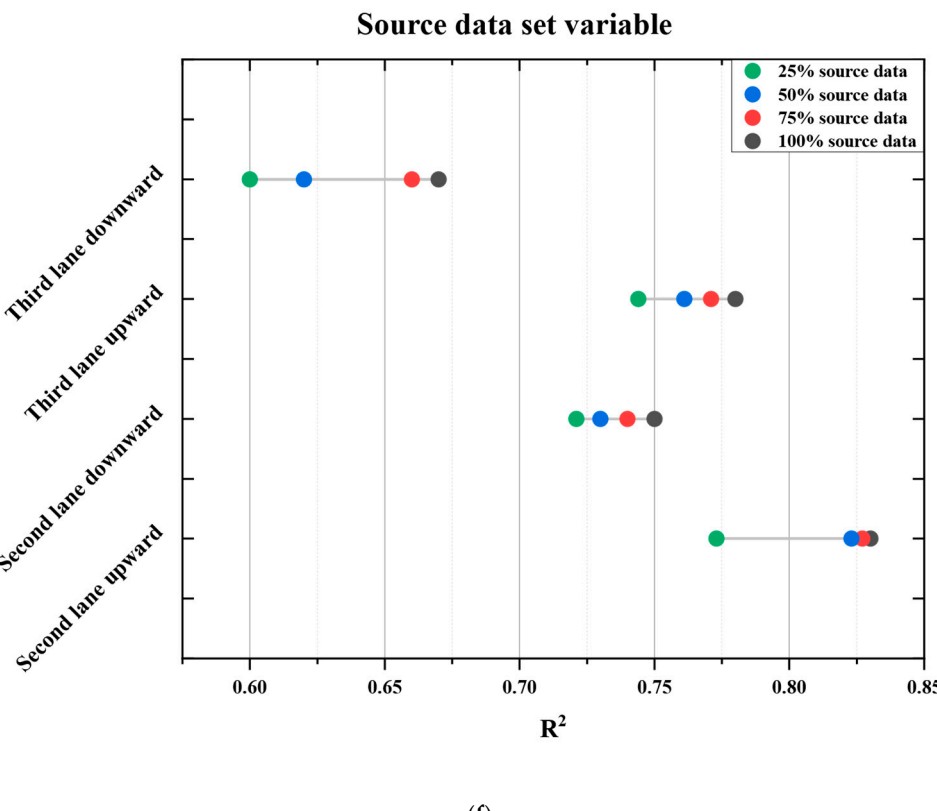

(**f**)

**Figure 12.** Impact of different dataset changes on the Two-stage TrAdaBoost.$R^2$ model. (**a**) Target dataset variable of RMSE; (**b**) target dataset variable of MAPE; (**c**) target dataset variable of $R^2$; (**d**) source dataset variable of RMSE; (**e**) source dataset variable of MAPE; (**f**) source dataset variable of $R^2$.

**Table 4.** Source domain dataset change.

| Lane | | RMSE | MAPE | $R^2$ |
|---|---|---|---|---|
| Second lane upward | 100% | 0.227 | 0.1107 | 0.83 |
| | 75% | 0.232 | 0.1144 | 0.82 |
| | 50% | 0.235 | 0.1174 | 0.82 |
| | 25% | 0.265 | 0.1833 | 0.77 |
| Second lane downward | 100% | 0.273 | 0.1208 | 0.75 |
| | 75% | 0.280 | 0.1255 | 0.74 |
| | 50% | 0.282 | 0.1281 | 0.73 |
| | 25% | 0.290 | 0.1305 | 0.72 |
| Third lane upward | 100% | 0.246 | 0.1001 | 0.78 |
| | 75% | 0.252 | 0.1067 | 0.77 |
| | 50% | 0.257 | 0.1061 | 0.76 |
| | 25% | 0.266 | 0.1299 | 0.74 |
| Third lane downward | 100% | 0.323 | 0.1037 | 0.67 |
| | 75% | 0.325 | 0.1045 | 0.66 |
| | 50% | 0.342 | 0.1568 | 0.62 |
| | 25% | 0.353 | 0.1673 | 0.60 |

**Table 5.** Target domain dataset change.

| Lane | | RMSE | MAPE | $R^2$ |
|---|---|---|---|---|
| Second lane upward | 100% | 0.227 | 0.1107 | 0.83 |
| | 75% | 0.239 | 0.1327 | 0.81 |
| | 50% | 0.259 | 0.1122 | 0.78 |
| | 25% | 0.279 | 0.1218 | 0.75 |
| Second lane downward | 100% | 0.273 | 0.1208 | 0.75 |
| | 75% | 0.310 | 0.1429 | 0.68 |
| | 50% | 0.325 | 0.1558 | 0.65 |
| | 25% | 0.352 | 0.1644 | 0.59 |
| Third lane upward | 100% | 0.246 | 0.1001 | 0.78 |
| | 75% | 0.252 | 0.1052 | 0.77 |
| | 50% | 0.261 | 0.1114 | 0.75 |
| | 25% | 0.278 | 0.1219 | 0.72 |
| Third lane downward | 100% | 0.323 | 0.1037 | 0.67 |
| | 75% | 0.332 | 0.1089 | 0.64 |
| | 50% | 0.350 | 0.1337 | 0.61 |
| | 25% | 0.365 | 0.154 | 0.57 |

Figure 12 shows that reducing the data in the source and target domains has different effects on the prediction accuracy of the transfer learning model. Taking the determination coefficient $R^2$ of the model as an example, regardless of changes in the size of the source or target domains, larger datasets all show better performance than smaller ones. Table 4 shows that for both the second and third lanes, predictive accuracy increases as the size of the source domain data increases. Table 5 shows a similar trend for the target domain datasets, showing that accuracy increases as the amount of target domain data is enhanced. This means that when dealing with new roads with small datasets, it is helpful to use available similar data and transfer learning methods to predict pavement performance. It also shows that the increasing range of $R^2$ when changing the target domain datasets is larger than that when changing the source domain datasets, as evidenced by comparing Tables 4 and 5. Other evaluation indexes follow the same trend, which means that the data from the target domain has a greater impact on the transfer learning model. Therefore, a higher percentage of target data does indeed help in obtaining better prediction accuracy of pavement usage performance.

The above introduction shows the great advantage of using transfer learning in pavement performance prediction. In practical engineering applications, however, the pavement management department needs to know the performance in advance at specific locations to make corresponding maintenance plans accordingly.

Figure 13 shows the predicted IRI and the corresponding measured IRI values along the mileposts of the road for the third lanes in both directions. The figure illustrates that the PSO-Two-stage TrAdaBoost.R$^2$ model has the best performance in following the trend of measured IRI. The PSO-Two-stage TrAdaBoost.R$^2$ model can precisely predict peak and valley values compared with the other three methods. The prediction results using only local data have the worst results, which means that prediction results with a small amount of data are inaccurate. These prediction results can offer direct guidance to the maintenance department for making plans at specific mileposts.

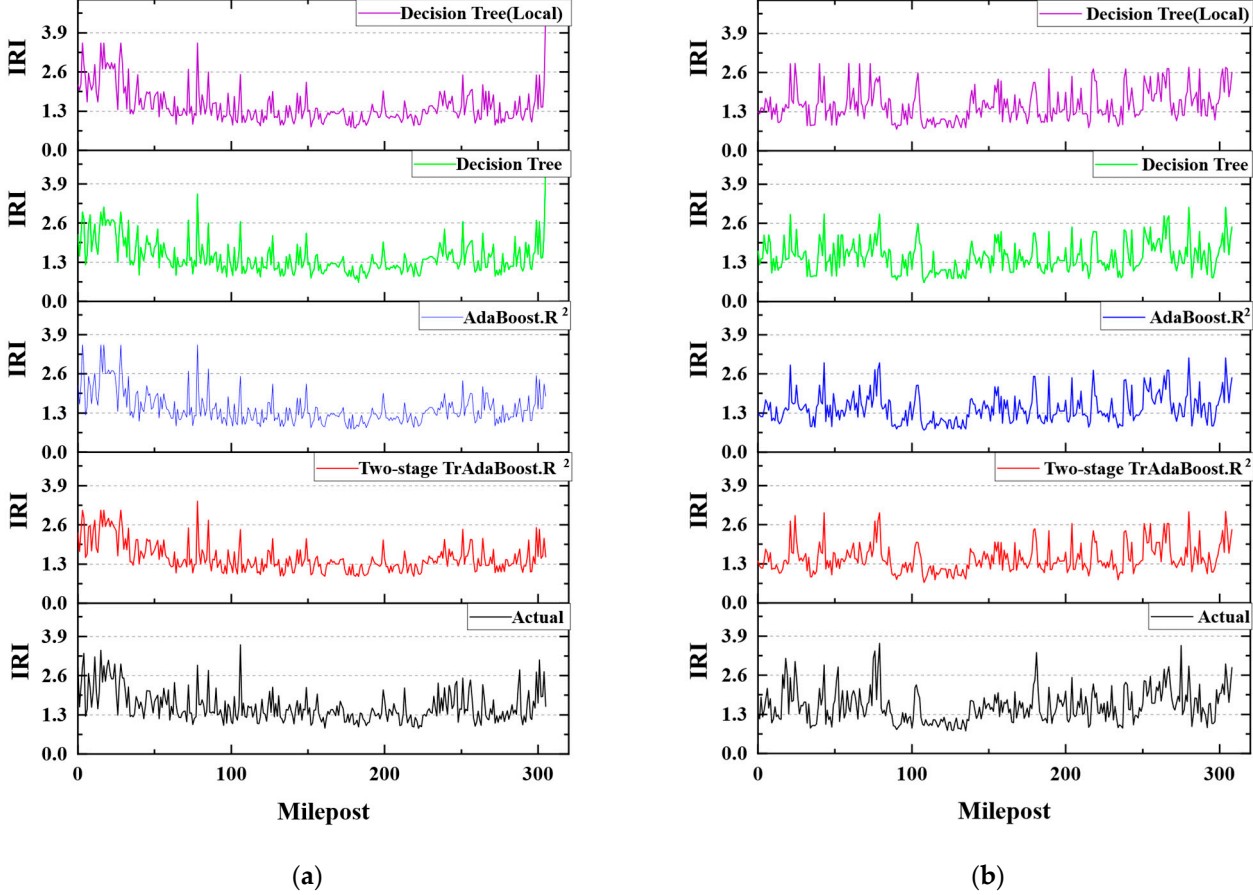

**Figure 13.** IRI prediction results along the milepost. (**a**) Distribution diagram of predicted IRI versus actual IRI of third lane upward; (**b**) distribution diagram of predicted IRI versus actual IRI of third lane downward.

## 6. Conclusions

Machine learning has been widely used in pavement performance prediction to help road management departments take preventive maintenance. Traditional machine learning method depend on large amounts of historical data for model training. This highly limits the application of machine learning in pavement preventive maintenance. This study provides an alternative method by training a deep learning model with insufficient data to enhance knowledge. The study, for the first time, developed a method that can use historical data from other roads to predict the performance of roads that lack data. The well-known open-source LTPP database is chosen as the source domain, and one of China's national highways, G18, is chosen as the target domain. The first three years of the G18's

historical data are used to simulate a newly built road. The instance-based transfer learning method Two-stage TrAdaBoost.$R^2$ algorithm, optimized with PSO, is proposed to predict the International Roughness Index of the road surface. The main contribution of the study are summarized as follows:

(1) Four different methods are compared, including the decision tree model trained only on local data, the decision tree model, AdaBoost.$R^2$ model, and the Two-stage TrAdaBoost.$R^2$ model trained using historical data from both local and open-source databases. The prediction results of the decision tree model using local data only and both local data and LTPP data yield $R^2$ values of 0.62 and 0.56, respectively. Although this shows that prediction accuracy could improve by enlarging the database, the prediction results still show low accuracy when using traditional machine learning methods.

(2) The proposed PSO-Two-stage TrAdaBoost.$R^2$ transfer learning method has better performance than the traditional machine learning method in predicting pavement performance. The average $R^2$ of the PSO-Two-stage TrAdaBoost.$R^2$ transfer learning method can reach 0.76 on average for all four lanes, which is 11% better than the AdaBoost.$R^2$ model and 22% better than the decision tree model. The best performance of $R^2$ is 0.83.

(3) The effects of the source domain and target domain are also examined in the study. Two groups of combinations are presented, including training the model using 100%, 75%, 50%, and 25% of data in the source domain, using all data in the target domain, and training the model using 100%, 75%, 50%, and 25% of data in the target domain and all data in the source domain. The results show that when predicting the performance of a new road with little dataset availability, it is helpful to use similar available data and transfer learning methods. It also shows that the increasing range of $R^2$ when changing target domain datasets is larger than that when changing source domain datasets.

In summary, this study has significant meaning for regions with no historical pavement performance data. This study used a large amount of historical road data from an open-source database and used the Two-stage TrAdaBoost.$R^2$ algorithm optimized with the PSO algorithm to predict the International Roughness Index of road surfaces, obtaining excellent prediction results. In addition, the method's robustness and stability are tested and verified across different lanes, obtaining good prediction results. Finally, the study explores the effect of varying the amount of data in different datasets on the model's prediction results. The experimental results show that changes in the amount of data in the target domain have a greater impact on model accuracy. This method could help these regions take advantage of the large amount of historical data from open-source databases. The local road management department could make a preventive maintenance plan based on the predicted results, thus saving the budget in the long term. In addition, the transfer learning approach used in this study could be extended to other engineering fields to enhance knowledge. Potential engineering fields include road safety performance functions, water quality prediction, air quality prediction, etc. However, further investigations need to be carried out to make the method more valuable in practical application. In this study, only the performance prediction of asphalt pavements was investigated, and further research is needed to address the performance prediction of cement concrete rigid pavements. Moreover, transfer learning from multiple source domains should be studied. The influence of features from the source domain should be further investigated, and more scenarios should be explored.

**Author Contributions:** Conceptualization, J.L.; methodology, J.G.; software, J.G.; resources, B.L.; writing—original draft preparation, J.G.; writing—review and editing, J.L.; supervision, B.L.; project administration, L.M. All authors have read and agreed to the published version of the manuscript.

**Funding:** This work was supported by the Natural Science Foundation of Hebei Province (No. E2022202007) and the Natural Science Foundation of Tianjin (22JCQNJC00400).

**Data Availability Statement:** The data presented in this study are available on request from the corresponding author. The data are not publicly available due to privacy.

**Conflicts of Interest:** Author Lingxin Meng was employed by the company Wenzhou Xinda Transportation Engineering Testing Co., Ltd. The remaining authors declare that the research was conducted in the absence of any commercial or financial relationships that could be construed as a potential conflict of interest.

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
