# Peer review of "Novel Instance-Based Transfer Learning for Asphalt Pavement Performance Prediction"

_buildings, doi:10.3390/buildings14030852_

Round 1

Reviewer 1 Report

Comments and Suggestions for Authors

In my opinion, the paper is almost ready to be published. The subject regarding the use of the technique presented (instanced-based transfer learning) on pavement performance prediction might help the pavement engineering area to improve design and maintenance activities. The results are well represented and discussed. The authors should carefully read the entire document because there a few grammar and writing issues. The main objectives of the paper should be highlighted in the abstract and in the introduction sections and compared to the findings shown in the conclusions section to check if these objectives were fully or partially reached. The conclusions are not supposed to summarize the main results and discussions. Also, the introduction section could be divided into two sections: introduction and review of literature. 

Comments on the Quality of English Language

The document must be carefully chcked for minor grammar and writing and spelling issues.

Reviewer 2 Report

Comments and Suggestions for Authors

An innovative particle swarm optimization (PSO) algorithm was proposed. The Long-Term Pavement Performance (LTPP) database is used as the source domain data and one of the highways in China is chosen as the target domain to predict the pavement performance. Results show that the proposed PSO-Two-stage TrAdaBoost.R2 model has the highest accuracy compared with AdaBoost.R2 model and traditional regression decision tree model. However, the introduction needs to be revised. The typographical errors need to be corrected. Paper content should be polished. The language also needs further revision. These issues must be addressed and require major revisions.

1.      Some abbreviations are not introduced at the first time.

2.      The introduction seems to be light and not rich enough. The research status of the DL-based methods is lacking. Better to supplement. Some key papers should be discussed:

a)      Automatic pixel-level detection of vertical cracks in asphalt pavement based on GPR investigation and improved mask R-CNN, https://doi.org/10.1016/j.autcon.2022.104689

b)      The use of deep neural networks for developing generic pavement rutting predictive models, https://doi.org/10.1080/10298436.2021.1942466

3.      Line 211: Is the title of the Figure 3 right?

4.      The grammar in the essay should be checked thoroughly.

5.      The overall quality of the pictures in this article is very poor. It is recommended to use vector images.

6.      Does your method work on all types of pavement? Is it robust?

7.      Limitations of the study should be appropriately mentioned in the conclusion.

8.      Authors should strictly follow the paper template on the official website of the conference for formatting.

9.      Conclusion: should be summarized and refined.

Comments on the Quality of English Language

Moderate editing of English language required

Reviewer 3 Report

Comments and Suggestions for Authors

good work

very minor comments, please check it.

thank you

Round 2

Reviewer 2 Report

Comments and Suggestions for Authors

The author has done a good job with the revisions, I have a few remaining further questions/remarks before publication.

1.       This manuscript should further add some articles and will be of interest to many, such as:

(1)   Feature extraction and segmentation of pavement distress using an improved hybrid task cascade network. International Journal of Pavement Engineering. DOI: 10.1080/10298436.2023.2266098.

(2)   Transfer learning for pavement performance prediction. International Journal of Pavement Research and Technology. DOI: 10.1007/s42947-019-0096-z.

2.       All figures should provide as clear a vector image as possible (It's best to use vector graphics).

3.       The template of the revised draft seems to be wrong, please refer to the latest template on the official website.
